# Application of a laser-based spectrometer for continuous insitu measurements of stable isotopes of soil CO₂ in calcareous and acidic soils

J. Joseph[1], C. Külls[2], M. Arend[3], M. Schaub[1], F. Hagedorn[1], A. Gessler[1] and M. Weiler[4]

[1] {Swiss Federal Institute for Forest, Snow and Landscape Research WSL, Zürcherstrasse 111, 8903 Birmensdorf, Switzerland}

[2] {Laboratory for Hydrology and International Water Management, University of Applied Sciences Lübeck, Germany}

[3] {Physiological Plant Ecology (PPE), Faculty of Integrative Biology, University of Basel, Switzerland}

[4] {Chair of Hydrology, Faculty of Environment and Natural resources, University of Freiburg, Germany}

Correspondence to: J. Joseph (jobin.joseph@wsl.ch)

**Abstract**
The short-term dynamics of carbon and water fluxes across the soil-plant-atmosphere continuum are still not fully
understood. One important constraint is the lack of methodologies that enable simultaneous measurements of soil
$CO_2$ concentration and respective isotopic composition at a high temporal resolution for longer periods of time.
$\delta^{13}C$ of soil $CO_2$ can be used to derive information on the origin and physiological history of carbon and $\delta^{18}O$ in
soil $CO_2$ aids to infer interaction between $CO_2$ and soil water. We established a real-time method for measuring
soil $CO_2$ concentration, $\delta^{13}C$ and $\delta^{18}O$ values across a soil profile at higher temporal resolutions (0.05 – 0.1 hz)
using an Off-Axis Integrated Cavity Output Spectrometer (OA-ICOS). We also developed a calibration method
correcting for the sensitivity of the device against concentration-dependent shifts in $\delta^{13}C$ and $\delta^{18}O$ values under
highly varying $CO_2$ concentration. The deviations of measured data were modelled, and a mathematical correction
model was developed and applied for correcting the shift. By coupling an OA-ICOS with hydrophobic but gas
permeable membranes placed at different depths in acidic and calcareous soils, we investigated the contribution of
abiotic and biotic components to total soil $CO_2$ release. We found that in the calcareous Gleysol, $CO_2$ originating
from carbonate dissolution contributed to the total soil $CO_2$ concentration at detectable degrees potentially due to
$CO_2$ evasion from groundwater. $^{13}C$-$CO_2$ of top soil at the calcareous soil site was found to be reflecting $\delta^{13}C$
values of atmospheric $CO_2$ and $\delta^{13}C$ of top soil $CO_2$ at the acidic soil site was representative of the biological
respiratory processes. $\delta^{18}O$ values of $CO_2$ in both sites reflected the $\delta^{18}O$ of soil water across most of the depth
profile, except for the 80 cm depth at the calcareous site where a relative enrichment in $^{18}O$ was observed.

**Key words:** $\delta^{13}C$, $\delta^{18}O$, OA-ICOS, hydrophobic/gas permeable membrane.

# 1    Introduction

Global fluxes of $CO_2$ and $H_2O$ are two major driving forces controlling earth's climatic systems. To understand the prevailing climatic conditions and predict climate change, accurate monitoring and modeling of these fluxes are essential (Barthel et al., 2014; Harwood et al., 1999; Schär et al., 2004). Soil respiration, the $CO_2$ flux released from soil surface to the atmosphere as a result of microbial and root respiration (heterotrophic and autotrophic) is the second largest terrestrial carbon flux (Bond-Lamberty and Thomson, 2010). The long-term dynamics of $CO_2$ release on a seasonal scale are reasonably well understood (Satakhun et al., 2013), whereas less information on $CO_2$ dynamics and isotopic composition are available for short-term variations on a diurnal scale (Werner and Gessler, 2011). The lack of proper understanding of the diurnal fluctuations in soil $CO_2$ release might introduce uncertainty in estimating the soil carbon budget and the $CO_2$ fluxes to the atmosphere. The isotopic composition of soil $CO_2$ and its diel fluctuation can be a critical parameter for the partitioning of ecosystem gas exchange into its components (Bowling et al., 2003; Mortazavi et al., 2004) and for disentangling plant and ecosystem processes (Werner and Gessler 2011). By assessing $\delta 1^3C$ of soil $CO_2$, it is possible to identify the source for $CO_2$ (Kuzyakov, 2006) and the coupling between photosynthesis and soil respiration when taking into account post-photosynthetic isotope fractionation (Werner et al., 2012; Wingate et al., 2010). $\delta^{13}C$ soil $CO_2$ reflects, however, not only microbial and root respiration but also abiotic sources from carbonate weathering (Schindlbacher et al., 2015).

Soil water imprints its $\delta^{18}O$ signature on soil $CO_2$ as a result of isotope exchange between $H_2O$ and $CO_2$ (aqueous). The oxygen isotopic exchange between $CO_2$ and soil water is catalyzed by microbial carbonic anhydrase (Sperber et al., 2015; Wingate et al., 2009). Thus, soil $CO_2$ can give information on the isotopic composition of both soil water resources and carbon sources. The oxygen isotope composition of plant-derived $CO_2$ is both, a tracer of photosynthetic and respiratory $CO_2$ and gives additional quantitative information on the water cycle in terrestrial ecosystems (Francey and Tans, 1987). To better interpret the $\delta^{13}C$ and $\delta^{18}O$ signals of atmospheric $CO_2$, the isotopic composition and its variability of the different sources need to be better understood (Werner et al., 2012; Wingate et al., 2010).

The conventional method to estimate $\delta^{13}C$ and $\delta^{18}O$ of soil $CO_2$ efflux is by using two end-member mixing models of atmospheric $CO_2$ and $CO_2$ produced in the soil (Keeling, 1958). The conventional methods for sampling soil produced $CO_2$ are chamber based (Bertolini et al., 2006; Torn et al., 2003), 'mini-tower' (Kayler et al., 2010; Mortazavi et al., 2004), and soil gas well (Breecker and Sharp, 2008; Oerter and Amundson, 2016) based methods. In conventional methods, air sampling is done at specific time intervals, and $\delta^{13}C$ and $\delta^{18}O$ are analyzed using Isotope Ratio Mass Spectrometry (IRMS) (Ohlsson et al., 2005). Such offline methods have several disadvantages like high sampling costs, excessive time consumption for sampling and analysis, increased sampling error and low temporal resolution. Kammer et al. (2011), showed how error-prone the conventional methods could be while calculating $\delta^{13}C$ and $\delta^{18}O$ (up to several per mil when using chamber and mini tower-based methods) (Kammer et al., 2011). In chamber-based systems, non-steady-state conditions may arise within the chamber due to increased $CO_2$ concentrations which in turn hinders the diffusion of $^{12}CO_2$ more strongly than that of heavier $^{13}CO_2$ (Risk and Kellman, 2008). Moreover, it has been found that $\delta^{18}O$ of $CO_2$ inside a chamber is significantly influenced by the $\delta^{18}O$ of the surface soil water as an equilibrium isotopic exchange happens during the upward diffusive movement of soil $CO_2$ (Mortazavi et al., 2004). The advent of laser-based isotope spectroscopy has enabled cost-effective, simple, and high precision real-time measurements of $\delta^{13}C$ and $\delta^{18}O$ in $CO_2$ (Kammer et al., 2011; Kerstel

and Gianfrani, 2008). This technique opened up new possibilities for faster and reliable measurements of stable
isotopes insitu, based on the principle of light absorption, using laser beams of distinct wavelengths in the near and
mid-infrared range (Bowling et al., 2003). Recently, several high frequency online measurements of $\delta^{13}C$ and $\delta^{18}O$
of soil $CO_2$ and $^2H$, $^{18}O$ of soil water vapor across soil depth profiles were reported by coupling either hydrophobic
but gas permeable membranes (installed at different depths in soil) or automated chamber systems with laser
spectrometers (Bowling et al., 2015; Jochheim et al., 2018; Stumpp et al., 2018). Such approaches enable detection
of vertical concentration profiles, temporal dynamics of soil $CO_2$ concentration and isotopic signature of soil $CO_2$
across different soil layers, thus aiding to identify and quantify various sources of $CO_2$ across the depth profile.
In 1988, O'Keefe and Decon introduced the Cavity Ring-Down Spectroscopy (CRDS) for measuring the isotopic
ratio of different gaseous species based on laser spectrometry (O'Keefe and Deacon, 1988). With the laser-based
spectrometry techniques, measuring sensitivities up to parts per trillion (ppt) concentrations are achieved (von
Basum et al., 2004; Peltola et al., 2012). In CRDS, the rate of change in the absorbed radiation of laser light that
is temporarily "trapped" within a highly reflective optical cavity is determined. This is achieved using resonant
coupling of a laser beam to the optical cavity and active locking of laser frequency to cavity length (Parameswaran
et al., 2009). Another well-established technique similar to CRDS is Off-Axis Integrated Cavity Output
Spectroscopy (OA-ICOS). It is based on directing lasers with narrowband and continuous-wave in an off-axis
configuration to the optical cavity (Baer et al., 2002).
Even though OA-ICOS can measure concentration and isotope signature of various gaseous species at high
temporal resolution, we found pronounced deviations in $\delta^{13}C$ and $\delta^{18}O$ measurements from the absolute values
when measured under changing $CO_2$ concentrations. So far to our knowledge, no study has been made available
detailing the calibration process of OA-ICOS $CO_2$ analyzers correcting for fluctuations of both $\delta^{13}C$ and $\delta^{18}O$
values under varying $CO_2$ concentrations. Most of the OA-ICOS $CO_2$ analyzers are built for working under stable
$CO_2$ concentrations, so that periodical calibration against in-house gas standards at a particular concentration is
sufficient. However, as there are pronounced gradients in $CO_2$ levels in soils (Maier and Schack-Kirchner, 2014),
$CO_2$ concentration depending shifts in measured isotopic values have to be addressed and corrected. Such
calibration is, however, also relevant for any other OA-ICOS application with varying levels of $CO_2$ (e.g., in
chamber measurements). Hence the first part of this work comprises the establishment of a calibration method for
OA-ICOS. The second part describes a method for online measurement of $CO_2$ concentrations and stable carbon
and oxygen isotope composition of $CO_2$ in different soil depths by coupling OA-ICOS with gas permeable
hydrophobic tubes (Membrane tubes, Accurel®). The use of these tubes for measuring soil $CO_2$ concentration (Gut
et al., 1998) and $\delta^{13}C$ of soil $CO_2$ (Parent et al., 2013) has already been established, but the coupling to an OA-
ICOS system has not been performed, yet.
We evaluated our measurement system by assessing and comparing the concentration, $\delta^{13}C$ and $\delta^{18}O$ of soil $CO_2$
for a calcareous and an acidic soil system. The primary foci of this study are to (1) introduce OA-ICOS in online
soil $CO_2$ concentration and isotopic measurements; (2) calibrate the OA-ICOS to render it usable for isotopic
analysis carried out under varying $CO_2$ concentrations; and (3) analyze the dynamics of $\delta^{13}C$ and $\delta^{18}O$ of soil $CO_2$
at different soil depths in different soil types at a higher temporal resolution.

## 2 Materials and Methods

### 2.1 Instrumentation

The concentration, $\delta^{13}C$ and $\delta^{18}O$ values of $CO_2$ were measured with an OA-ICOS, as described in detail by Baer et al. (2002) and Jost et al. (2006). In this study, we used an OA-ICOS, (LGR-CCIA 36-d) manufactured by Los Gatos Research Ltd, San-Francisco, USA. LGR-CCIA 36-d measures $CO_2$ concentration, and $\delta^{13}C$ and $\delta^{18}O$ values at a frequency up to 1 Hz. The operational $CO_2$ concentration range was 400 to 25,000 ppm. Operating temperature range was +10 - +35°C, and sample temperature range (Gas temperature) was between -20°C and 50°C. Recommended inlet pressure was < 0.0689 MPa. The multiport inlet unit, an optional design that comes along with LGR-CCIA 36-d, had a manifold of 8 digitally controlled inlet ports and one outlet port. It rendered the user with an option of measuring eight different $CO_2$ samples at the desired time interval. Three standard gases with distinct $\delta^{13}C$ and $\delta^{18}O$ values were used for calibration in this study (See Supplementary Table.1). The standard gases used in this study were analyzed for absolute concentration and respective $\delta^{13}C$ and $\delta^{18}O$ values. $\delta$-values are expressed based on Vienna Pee Dee Belemnite (VPDB)-$CO_2$ scale, and were determined by high precision IRMS analysis.

### 2.2 Calibration setup and protocol

We developed a two-step calibration procedure to; a) correct for concentration-dependent errors in isotopic data measurements, and b) correct for deviations in measured $\delta$-values from absolute values due to offset (other than concentration-dependent error) introduced by the laser spectrometer. Also, we used Allan variance curves for determining the time interval to average the data (Nelson et al., 2008) to achieve the highest precision that can be offered by LGR-CCIA 36-d (Allan et al., 1997).

The first part of our calibration methodology was developed to correct for the concentration-dependent error observed in preliminary studies for $\delta^{13}C$ and $\delta^{18}O$ values measured using OA-ICOS. Such a calibration protocol was used in addition to the routine three-point calibration performed with in-house $CO_2$ gas standards of known $\delta^{13}C$ and $\delta^{18}O$ values. We developed a $CO_2$ dilution set up (See Figure. 1), with which each of the three $CO_2$ standard gases was diluted with synthetic $CO_2$ free air (synth-air) to different $CO_2$ concentrations. By applying a dilution series, we identified the deviation of the measured (OA-ICOS) from the absolute (IRMS) $\delta^{13}C$ and $\delta^{18}O$ values depending on $CO_2$ concentration (See Figure.4). The $\delta^{13}C$ and $\delta^{18}O$ values of our inhouse calibration gas standards were measured via cryo-extraction and Dual Inlet IRMS. $\delta^{13}C$, and $\delta^{18}O$ of the standard gases (See Supplementary Table.1) across a wide range of $CO_2$ concentrations are measured using OA-ICOS. The deviation of the measured $\delta^{13}C$, and $\delta^{18}O$ from absolute values with respect to changing $CO_2$ concentrations was mathematically modeled and later used for data correction (See Figure.5). A standard three-point calibration was then applied correcting for concentration-dependent errors (See Figure.7). The standards used covered a wide range of $\delta^{13}C$ and $\delta^{18}O$, including the values observed in the field of application.

Standard gases were released to a mass flow controller (ANALYT-MTC, series 358, MFC1) after passing through a pressure controller valve (See Figure. 1) with safety bypass (TESCOM, D43376-AR-00-X1-S; V5). A Swagelok

filter, ((Stainless Steel All-Welded In-Line Filter (Swagelok, SS-4FWS-05; F1)) was installed at the inlet of the flow controller (ANALYT-MTC, series 358; MFC1). Synth-air was released and passed to another flow controller (ANALYT-MTC, series 358; MFC2) through a Swagelok filter (F2 in Figure. 1). $CO_2$ and synth-air leaving the flow controllers (MFC1 and MFC2 respectively) were then mixed and drawn through a 6.35 mm outer diameter (OD) Teflon tube (P8), which was kept in a gas thermostat unit (See Figure.1). The thermostat unit contained, a) a thermostat-controlled water bath (Kottermann, 3082) and b) an Isotherm flask containing liquid nitrogen. The water bath was used to raise the temperature above room temperature and also to bring the temperature down to +5°C, by placing ice packs in the water bath. To reach low temperatures (-20°C), we immersed the tubes in the isotherm flask filled with liquid $N_2$. Leaving the thermostat unit, the gas was directed to the multiport inlet unit of the OA-ICOS. By using the thermostat unit, we introduced a shift in the reference gas temperature and the aim was to test the temperature sensitivity of the OA-ICOS in measuring $\delta^{13}C$ and $\delta^{18}O$ values. The third $CO_2$ standard gas (which is used for validation) was produced by mixing the other two gas standards in equal molar proportions in a 10L volume plastic bag with inner aluminum foil coating and welded seams ($CO_2$ mix: Linde PLASTIGAS®) under 0.03 MPa pressure by diluting to the required concentration using synth-air. The mixture was then temperature adjusted and delivered to the multiport inlet unit (MIU) by using a 6.35 mm (OD) Teflon tube (P10). From the multiport inlet unit, calibration gases were delivered into the OA-ICOS for measurement using a 6.35 mm (OD) Teflon tube (P9) at a pressure < 0.0689 MPa, with a flow rate of 500 mL/min. The gas leaving the OA-ICOS through the exhaust was fed back to the 6.35 mm (OD) Teflon tube (P8) by using a Swagelok pipe Tee (Stainless Steel Pipe Fitting, Male Tee, 6.35 mm (OD). Male NPT), intersecting P8 line before entering the thermostat unit. Thus, the gas fed was looped in the system until steady values were reported by the OA-ICOS based on $CO_2$ [ppm], $\delta^{13}C$ and $\delta^{18}O$ measurements. $CO_2$ gas standards were measured at 27 different $CO_2$ concentration levels ranging between 400 and 25,000 ppm. Every hour before sampling, synth-air gas was flushed through the system to remove $CO_2$ to avoid memory effects. The calibration gases were measured in a sequence one after the other four times. During each round of measurement, every calibration gas was diluted to different concentrations of $CO_2$ (400 - 25,000 ppm) and the respective isotopic signature and concentration were determined. For each measurement of $\delta^{13}C$ and $\delta^{18}O$ at a given concentration, the first 50 readings were omitted to avoid possible memory effects of the laser spectrometer and the subsequent readings for the next 256 seconds were taken and averaged to get maximum precision for $\delta^{13}C$ and $\delta^{18}O$ measurements. When switching between different calibration gases at the multiport inlet unit, synth-air was purged through the systems for 30 seconds to avoid cross-contamination.

## 2.3 Experimental Sites

*In situ* experiments were conducted to measure $\delta^{13}C$, $\delta^{18}O$ and concentrations of soil $CO_2$ in two different soil types (calcareous and acidic soil). The measurements in a calcareous soil were conducted during June 2014 in cropland cultivated with wheat (*Triticum aestivum*) in Neuried, a small village in the Upper Rhine Valley in Germany situated at 48°26'55.5"N, 7°47'20.7"E, 150 m a.s.l. The soil type described as calcareous fluvic Gleysol IUSS (2014) developed on gravel deposits in the upper Rhine valley. Soil depth was medium to deep, with high contents of coarse material (> 2 mm) up to 30 - 50%. Mean soil organic carbon (SOC) content was 1.2 - 2% and,

SOC stock was ranging between 50 and 90 t/ha. The average pH was found to be 8.6. The study site receives an annual rainfall of 810 mm and has a mean annual temperature of 12.1°C.

In situ measurements in an acidic soil were conducted by the end of July 2014 in the model ecosystem facility (MODOEK) of the Swiss Federal Research Institute WSL, Birmensdorf, Switzerland (47°21'48" N, 8°27'23" E, 545 m a.s.l.). The MODOEK facility comprises 16 model ecosystems, belowground split into two lysimeters with an area of 3 m$^2$ and a depth of 150 cm. The lysimeters used for the present study were filled with acidic (haplic Alisol) forest soil IUSS (2014) and planted with young beech trees (Arend et al., 2016). The soil pH was 4.0 and a total SOC content of 0.8% (Kuster et al., 2013).

## 2.4 Experimental Setup

The OA-ICOS was connected to gas permeable, hydrophobic membrane tubes (Accurel® tubings, 8 mm OD) of 2 m length, placed horizontally in the soil at different depths. Tubes were laid in six different depths (4, 8, 12, 17, 35, and 80 cm) for calcareous soil and three (10, 30, and 60 cm) for acidic soil.

Technical details of the measurement setup are shown in Figure 2. Both ends of the membrane tubes were extended vertically upwards reaching the soil top by connecting them to gas impermeable Synflex® tubings (8 mm OD) using Swagelok tube fitting union (Swagelok: SS-8M0-6, 8 mm Tube OD). One end of the tubing system was connected to a solenoid switching valve (Bibus: MX-758.8E3C3KK) and by using a stainless-steel reducing union (Swagelok: SS-8M0-6-6M), to the outlet of the LGR CCIA 36-d by using 6.35 mm (OD) Teflon tubing. The other end was connected via the multiport inlet unit to the gas inlet of the LGR CCIA 36-d.

This way, a loop was created in which the soil $CO_2$ drawn into the OA-ICOS was circulated back through the tubes and in and out of the OA-ICOS and measured until a steady state was reached. We experienced no drop in cavity pressure while maintaining a closed loop (See Supplementary Figure S2). Each depth was selected and continuously measured for 6 minutes at specified time intervals by switching to defined depths at the multiport inlet unit and also at the solenoid valve.

## 3 Results and Discussion

### 3.1 Instrument calibration and correction

The highest level of precision obtained for $\delta^{13}C$ and $\delta^{18}O$ measurements at the maximum measuring frequency (1Hz) were determined by using Allan deviation curves (see Figure 3). Maximum precision of 0.022‰ for $\delta^{13}C$ was obtained when the data were averaged over 256 seconds, and for $\delta^{18}O$, 0.077‰ for the same averaging interval as for $\delta^{13}C$.

To correct for $CO_2$ concentration-dependent errors in raw $\delta^{13}C$ and $\delta^{18}O$ data, we analysed data obtained from the OA-ICOS to determine the sensitivity of $\delta^{13}C$ and $\delta^{18}O$ measurements against changing concentrations of $CO_2$. We observed a specific pattern of deviance in the measured isotopic data from the absolute values (both for $\delta^{13}C$ and

$\delta^{18}O$) across $CO_2$ concentration ranging from 25,000 to 400 ppm (See Figure.4). Uncalibrated $\delta^{13}C$ and $\delta^{18}O$
measurements showed a standard deviation of 6.44 ‰ and 6.80 ‰ respectively, when measured under changing
$CO_2$ concentrations.
The dependency of $\delta^{13}C$ and $\delta^{18}O$ values on the $CO_2$ concentration was compensated by using a nonlinear model.
The deviations (Diff-$\delta$) of the measured delta values ($\delta_{(OA\text{-}ICOS)}$) from the absolute value of the standard gas ($\delta$
$_{(IRMS)}$) at different concentrations of $CO_2$ were calculated (Diff-$\delta$ = $\delta_{(OA\text{-}ICOS)}$ - $\delta_{(IRMS)}$). Several mathematical
models were then fitted on Diff-$\delta$ as a function of changing $CO_2$ concentration (See figure.5). The mathematical
model with the best fit for Diff-$\delta$ data was selected using Akaike information criterion corrected (AICc) (Glatting
et al., 2007; Hurvich and Tsai, 1989; Yamaoka et al., 1978). The non-linear model fits applied for Diff-$\delta^{13}C$, and
Diff-$\delta^{18}O$ measurements are given in Tables 1 & 2, respectively. For Diff-$\delta^{13}C$, a three-parameter exponential
model fitted best with $r^2 = 0.99$ (see Table 3 for the values of the parameters, see supplementary Figure S3 (a) for
model residuals), and a three-parameter power function model (see Table 2) with $r^2 = 0.99$ showed the best fit for
Diff-$\delta^{18}O$ (see Table 3 for the values of the parameters, see supplementary Figure S3 (b) for model residuals). The
best fit was then introduced into the measured isotopic data ($\delta^{13}C$ and $\delta^{18}O$) and corrected for concentration-
dependent errors (See figure. 6). After correction, the standard deviation of $\delta^{13}C$ was reduced to 0.08 ‰ and of
$\delta^{18}O$ to 0.09 ‰ for all measurements across the whole $CO_2$ concentration range.

After correcting the measured $\delta^{13}C$ and $\delta^{18}O$ values for the $CO_2$ concentration-dependent deviations, a three-point
calibration (Sturm et al., 2012) was made by generating linear regressions with the concentration corrected $\delta^{13}C$
and $\delta^{18}O$ values against absolute $\delta^{13}C$ and $\delta^{18}O$ values (See Figure.7, see supplementary Figure S4 for linear
regression residuals). Using the linear regression lines, we were able to measure the validation gas standard with
standard deviations of 0.0826 ‰ for $\delta^{13}C$ and 0.0941 ‰ for $\delta^{18}O$.
For the LGR CCIA 36-d, we found that routine calibration (Correction for concentration-dependent error plus
three-point calibration) was necessary for obtaining the required accuracy, in particular under fluctuating $CO_2$
concentrations. The LGR CCIA-36d offers an option for calibration against a single standard, a feature which was
already in place in a predecessor model (CCIA DLT-100) (Guillon et al., 2012). This internal calibration is
sufficient, when LGR CCIA-36d is operated only under stable $CO_2$ concentrations. To correct for the concentration
dependency, we introduced mathematical model fits, which corrected for the deviation pattern found for both $\delta^{13}C$
and $\delta^{18}O$. We assume that these deviations are instrument specific and the fitting parameters need to be adjusted
for every single device. Experiments conducted to investigate the influence of external temperature fluctuations
on OA-ICOS measurements did not show any significant changes in the temperature inside the optical cavity of
OA-ICOS (See Supplementary Figure S1). The previous version of the Los Gatos CCIA was strongly influenced
by temperature fluctuations during sampling (Guillon et al., 2012). The lack of temperature dependency as
observed here with the most recent model can be mostly due to the heavy insulation provided with the system,
which was not found in the older models.
Guillon et al. (2012) found a linear correlation between $CO_2$ concentration and respective stable isotope signatures
with a previous version of the Los Gatos CCIA $CO_2$ stable isotope analyser. In our experiments with the OA-ICOS,
best fitting correlation between $CO_2$ concentration and $\delta^{13}C$ and $\delta^{18}O$ measurements were exponential and power

functions, respectively. We assume that measurement accuracy is influenced by the number of $CO_2$ molecules present inside the laser cavity of the particular laser spectrometer as we observed large standard deviation in isotopic measurements at lower $CO_2$ concentrations. This behavior of an OA-ICOS can be expected as it functions by sweeping the laser along an absorption spectrum, measuring the energy transmitted after passing through the sample. Therefore, energy transmitted is proportional to the gas concentration in the cavity. The laser absorbance is then determined by normalizing against a reference signal, finally calculating the concentration of the sample measured by integrating the whole spectrum of absorbance (O'Keefe et al., 1999).

## 3.2    Variation in soil $CO_2$ concentration, carbon and oxygen isotope values

Figures 9 and 10 show the $CO_2$ concentration, $\delta^{13}C$ and $\delta^{18}O$ measurements of soil $CO_2$ in the calcareous as well as in the acidic soil across the soil profile with sub-daily resolution and as averages for the day, respectively. We observed an increase in the $CO_2$ concentration across the soil depth profile for both, the calcareous and the acidic soil. Moreover, there were rather contrasting $\delta^{13}C$ values across the profile for the two soil types. In the calcareous soil, $CO_2$ was relatively enriched in $^{13}C$ in the surface soil (4 cm) as compared to the 8 cm depth. Below 8 cm down to 80 cm depth, we found an increase in $\delta^{13}C$ values. At 80 cm depth, the $\delta^{13}C$ in soil $CO_2$ ranged between -7.15 and -3.35 ‰ (See Figure. 9) with a daily average of -6.19 ± 1.45 ‰ (See Figure. 10) and hence clearly above atmospheric values ($\approx$ -8.0 ‰). For $\delta^{18}O$ values of calcareous soil, the depth profile showed no specific pattern except for the $\delta^{18}O$ values at 80 cm depth was found to be less negative than the values of the other depths. The $\delta^{18}O$ value in the top 4 cm was found to be slightly more enriched that the 8 cm depth and between 8 cm – 35 cm, $\delta^{18}O$ values showed little variation relative to each other. For the sub-daily measurements, we observed a sharp decline in $\delta^{18}O$ values at around 02:00, which is also observed but less pronounced for $\delta^{13}C$ signal. We assume that, the reason for such aberrant values is rather a technical issue than a biological process. It could be due to the fact that the internal pump in the OA-ICOS was not taking adequate amount of gas into the optical cavity, thereby creating a negative pressure inside the cavity resulting in the observed aberrant values. The patterns observed for the $\delta^{13}C$ values of $CO_2$ in the calcareous soil with $^{13}C$ enrichment in deeper soil layers can be explained by a substantial contribution of $CO_2$ from abiotic origin to total soil $CO_2$ release as a result of carbonate weathering and subsequent out-gassing from soil water (Schindlbacher et al., 2015). According to Cerling (1984), the distinct oxygen and carbon isotopic composition of soil carbonate depends primarily on the isotopic signature of meteoric water and to the proportion of $C_4$ biomass present at the time of carbonate formation (Cerling, 1984), but also on numerous other factors that determine the $^{13}C$ value of soil $CO_2$. $CO_2$ released as a result from carbonates in calcareous soil site have a distinct $\delta^{13}C$ value of -9.3 (mean value across soil profile 0 - 80 cm depth) (Figure 8(c)), while $CO_2$ released during biological respiratory processes has $\delta^{13}C$ values around -24‰ as observed in the acidic soil (Figure 10 (e)). The $\delta^{13}C$ values of soil $CO_2$ observed in the deepest soil layer in the calcareous soil site most likely indicates the presence of carbonate sources of pedogenic and geologic origin. Even though the contribution of $CO_2$ from abiotic sources to soil $CO_2$ is often considered to be low, several studies have reported significant proportions ranging between (10 - 60%) emanating from abiotic sources (Emmerich, 2003; Plestenjak et al., 2012; Ramnarine et al., 2012; Serrano-Ortiz et al., 2010; Stevenson and Verburg, 2006; Tamir et al., 2011). Bowen and Beerling, (2004) showed that isotope effects associated with soil organic matter decomposition can cause a strong gradient in $\delta$ values of soil organic matter (SOM) with depth, but are not always reflected in the $\delta^{13}C$ values of soil

$CO_2$. We have measured soil samples for bulk soil $\delta^{13}C$, carbonate $\delta^{13}C$ & $\delta^{18}O$ values and also determined the
percentage of total carbon in the soil across a depth profile of (0-80 cm) (See Figure 8). We observed an increase
in $\delta^{13}C$ values for bulk soil in deeper soil layers (See Figure 8 (a,c)). Moreover, also the carbonate $\delta^{13}C$ values got
more positive in the 60-80 cm layer. Since total organic carbon content decreases with depth it can be assumed
that $CO_2$ derived from carbonate weathering having less negative $\delta^{13}C$ more strongly contributed to the soil $CO_2$
(especially since we see an increase in soil $CO_2$ concentration with depth). This is accordance with the laser-based
measurements which showed a strong increase in $\delta^{13}C$ of soil $CO_2$ in the deepest soil layer leading us to the
hypothesis that this signal is indicating a strong contribution of carbonate derived $CO_2$. Water content, soil $CO_2$
concentration and presence of organic acids or any other source of $H^+$ are the major factors influencing carbonate
weathering, and variations in soil $CO_2$ partial pressure, moisture, temperature, and pH can cause degassing of $CO_2$
which contributes to the soil $CO_2$ efflux (Schindlbacher et al., 2015; Zamanian et al., 2016). $CaCO_3$ solubility in
pure $H_2O$ at 25°C is 0.013 $gL^{-1}$, but in weak acids like carbonic acid, the solubility is increased up to five fold
(Zamanian et al., 2016). The production of carbonic acid due to $CO_2$ dissolution will convert carbonate to
bicarbonates resulting in exchange of carbon atoms between carbonates and dissolved $CO_2$. We assume that at our
study site, the topsoil is de-carbonated due to intensive agriculture for a longer period and thus soil $CO_2$ there
originates primarily from autotrophic and heterotrophic respiratory activity. In contrast to the deeper soil layers,
where the carbonate content is high, $CO_2$ from carbonate weathering is assumed to be a dominating source of soil
$CO_2$. Also, outgassing of $CO_2$ from the large groundwater body underneath the calcareous Gleysol might contribute
to the inorganic $CO_2$ sources in the deeper soil as we found ground water table to be 1-2m below the soil surface.
Relative $^{13}C$ enrichment of the $CO_2$ in the topsoil (4 cm) compared to 8 cm depth is probably due to the invasive
diffusion of atmospheric $CO_2$ which has a $\delta^{13}C$ value close to -8‰ (e.g., (Levin et al., 1995) ). The $\delta^{18}O$ patterns
for $CO_2$ between 4 and 35 cm might reflect the $\delta^{18}O$ of soil water with stronger evaporative enrichment at the top
and $^{18}O$ depletion towards deeper soil layers. In comparison, the strong $^{18}O$ enrichment of soil $CO_2$ towards 80 cm
in the calcareous Gleysol very likely reflects the $^{18}O$ values of groundwater lending further support for the high
contribution of $CO_2$ originating from the outgassing of groundwater. We, however, need then to assume that that
the oxygen in the $CO_2$ is not in full equilibrium with the precipitation influenced soil water. As mainly microbial
carbonic anhydrase mediates the fast equilibrium between $CO_2$ and water in the soil and the microbial activity is
low in deeper soil layers (Schmidt et al., 2011), we speculate that in deep layers with a significant contribution of
ground-water derived $CO_2$ to the $CO_2$ pool, a lack of full equilibrium with soil water might be the reason for the
observed $\delta^{18}O$ values.

Soil $CO_2$ concentration in the acidic soil showed a positive relationship with soil depth as $CO_2$ concentration
increased along with increasing soil depth (Figs. 9 & 10). $CO_2$ concentrations were distinctly higher than in the
calcareous soil, very likely due to the finer texture than in the gravel-rich calcareous soil. $\delta^{13}C$ values amounted to
approx. – 26 ‰ in 30 and 60 cm depth indicating the biotic origin from (autotrophic and heterotrophic) soil
respiration (Schönwitz et al., 1986). In the topsoil, $\delta^{13}C$ values did not strongly increase, pointing towards a less
pronounced inward diffusion of $CO_2$ in the acidic soil site, most likely due to more extensive outward diffusion of
soil $CO_2$ as indicated by the still very high $CO_2$ concentration at 10 cm creating a sharp gradient between soil and
atmosphere. Moreover, the acidic soil was rather dense and contained no stones, strongly suggesting that gas
diffusivity was rather small. $\delta^{18}O$ depths patterns of soil $CO_2$ in the acidic soil were most likely reflecting $\delta^{18}O$
values of soil water as $CO_2$ became increasingly $^{18}O$ depleted from top to bottom. $\delta^{18}O$ of deeper soil layers $CO_2$
(30 - 60 cm) was close to the values expected when full oxygen exchange between soil water and $CO_2$ occurred
(Kato et al., 2004). Assuming an $^{18}O$ fractionation of 41‰ between $CO_2$ and water (Brenninkmeijer et al., 1983)
this would result in an expected value for $CO_2$ of $\approx$ -10 ± 2‰ vs. VPDB-$CO_2$. Corresponding results had been
shown for $\delta^{18}O$ of soil $CO_2$ using similar hydrophobic gas permeable membrane tubes used when measuring $\delta^{18}O$
of soil $CO_2$ and soil water *in situ* (Gangi et al., 2015).
**4    Conclusions**
During our preliminary tests with the OA-ICOS, we found that the equipment was highly sensitive to changes in
$CO_2$ concentrations. We found a non-linear response of the $\delta^{13}C$ and $\delta^{18}O$ values against changes in $CO_2$
concentration. Given the fact that laser-based $CO_2$ isotope analyzers are deployed on site in combination with
different gas sampling methods like automated chambers systems (Bowling et al., 2015), and hydrophobic gas
permeable membranes (Jochheim et al., 2018) for tracing various ecosystem processes, it is important to address
this issue. Therefore, we developed a calibration strategy for correcting errors introduced in $\delta^{13}C$ and $\delta^{18}O$
measurements due to the sensitivity of the device against changing $CO_2$ concentrations.  We found that the OA-
ICOS measures stable isotopes of $CO_2$ gas samples with a precision comparable to conventional IRMS.  The
method described in this work for measuring $CO_2$ concentration, $\delta^{13}C$ and $\delta^{18}O$ values in soil air profiles using an
OA-ICOS and hydrophobic gas permeable tubes are promising and can be applied for soil $CO_2$ flux studies. As
this set up is capable of measuring continuously for longer time periods at higher temporal resolution ($0.05 - 0.1$
Hz), it offers greater potential to investigate the isotopic identity of $CO_2$ and the interrelation between soil $CO_2$ and
soil water. By using our measurement setup, we could identify abiotic as well as biotic contributions to the soil
$CO_2$ in the calcareous soil. We infer that that degassing of $CO_2$ from carbonates due to weathering and evasion of
$CO_2$ from groundwater may leave the soil $CO_2$ with a specific and distinct $\delta^{13}C$ signature especially when the biotic
activity is rather low.


**Acknowledgements**
We thank Federal Ministry of Education and Research, Germany (BMBF), KIT (Karlsruhe Institute of
Technology) for providing financial support for the project ENABLE-WCM (Grant Number: 02WQ1205). AG
and JJ acknowledge financial support by the Swiss National Science Foundation (SNF; 31003A_159866). We
thank Barbara Herbstritt, Hannes Leistert , Emil Blattmann and Jens Lange, Matthias Saurer, Alessandro Schlumpf,
Lukas Bächli and Christian Poll for outstanding support in getting this project into a reality.

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

560

Table 1. Correction factor models are fitted for Diff-$\delta^{13}$C, DF (Degrees of Freedom), AIC$_C$
(Akaike information criterion) and [CO$_2$] CO$_2$ concentration in ppm

| Model Fit | Equation | $R^2$ | AIC$_C$ | DF |
|---|---|---|---|---|
| Exponential | $Diff - \delta 13C = a * (b - \exp(-c * [CO2]))$ | 0.99 | -294.6 | 54 |
| Polynomial | $Diff - \delta 13C = a + b * [CO2] + c/[CO2]\^2)$ | 0.98 | -27.56 | 54 |
| Logarithmic | $Diff - \delta 13C = a + b * ln([CO2])$ | 0.89 | 91.68 | 55 |
| Lowess | ------ | 0.99 | -170.24 | 54 |

Table 2. Correction factor models are fitted for Diff-$\delta^{18}$O, DF (Degrees of Freedom), AIC$_C$

(Akaike information criterion) and [CO$_2$] CO$_2$ concentration in ppm.

| Model Fit | Equation | R$^2$ | AIC$_C$ | DF |
|-----------|----------|-------|---------|-----|
| Power | $Diff - \delta 18O = a * \left(b^{[CO2]}\right) * ([CO2]^c)$ | 0.99 | -337.04 | 51 |
| Polynomial | $Diff - \delta 18O = (a + b * x)/(1 + c * [CO2] + d * [CO2]\,^{\wedge}2)$ | 0.98 | -19.34 | 50 |
| Stein-Hart | $Diff - \delta 18O = 1/a + (b * \ln[CO2]) + (c * (ln[CO2])^3)$ | 0.96 | 29.77 | 51 |
| Lowess | ------- | 0.78 | 128.66 | 51 |

Table 3. Parameter values for correction factor model fit for Diff-$\delta^{13}$C & Diff-$\delta^{18}$O.

| Parameter | Value | Std Error | 95% Confidence |
|---|---|---|---|
| $a^{13}$C | 31.007 | 0.2149 | 30.57 - 31.43 |
| $b^{13}$C | 0.713 | 0.002376 | 0.708995 - 0.718522 |
| $c^{13}$C | 0.000043 | 0.000000 | 0.000042 - 0.000043 |
| $a^{18}$O | 0.85 | 0.003 | 0.8455 – 0.8576 |
| $b^{18}$O | 0.99 | 0.00 | 0.999928 – 0.9999283 |
| $c^{18}$O | 0.477 | 0.0047 | 0.476871 – 0.478767 |

# Figure 1

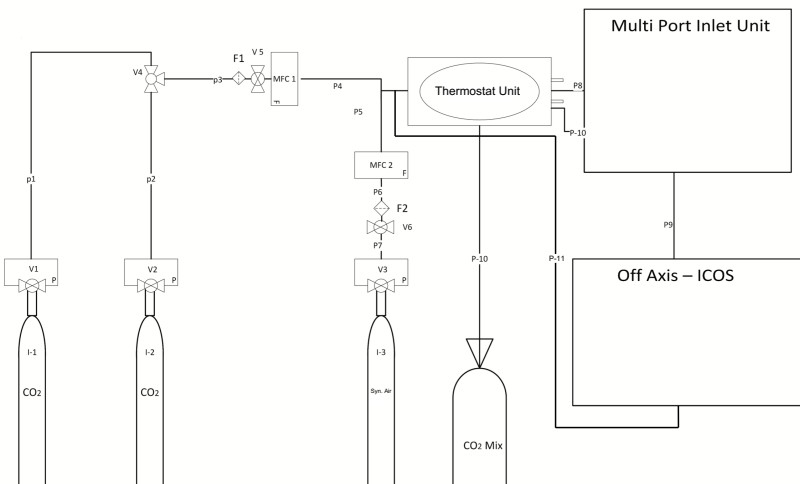

**Figure 1**: Setup made for calibration of OA- ICOS (LGR-CCIA 36-d). I(1,2): CO2 standards, CO2 Mix: Gas standards mixed in equal molar proportion, I3: Synthetic Air, MFC(1, 2): Mass Flow Controller, F(1, 2): PTFE filter, V(1, 2, 3): Pressure reducing Valves, V4: Three way ball valve, V(5,6): pressure controller valve with safety bypass , P (1-7): Steel pipes, P(8-11):Teflon tubing.

**Figure 2**

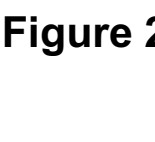

OA-ICOS

outlet        inlet

Air flow direction

Solenoid Valve        MIU

cm
cm
cm

17 cm

35 cm

80 cm

**Figure 2**: Installation made for soil air $CO_2$ [ppm], $\delta^{13}C$-$CO_2$ and $\delta^{18}O$ -$CO_2$ measurements using an Off-Axis integrated cavity output spectrometer (OA-ICOS). Hydrophobic membrane tubing were installed horizontally in soil at different depths. MIU: multi-port inlet unit

# Figure 3

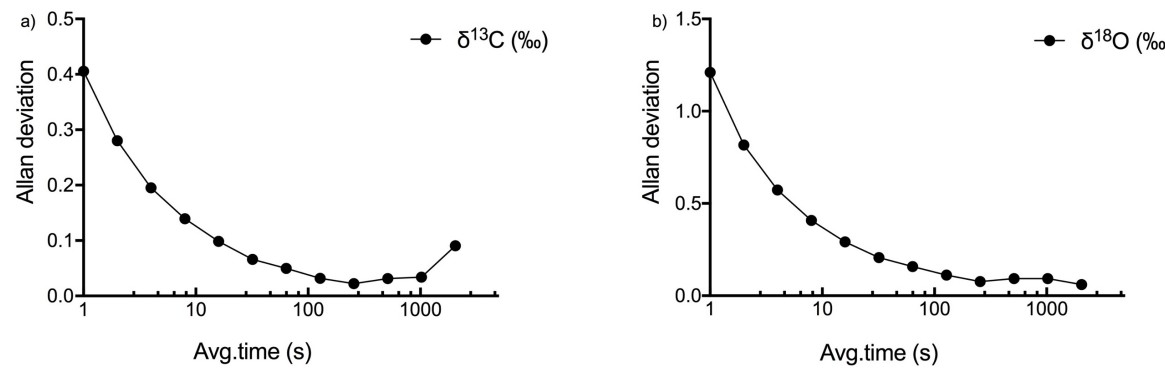

**Figure 3**: Allan deviation curve for δ¹³C (a) and δ¹⁸O(b) measurements by OA-ICOS $CO_2$ Carbon isotope analyzer (LGR CCIA-36d).

# Figure 4

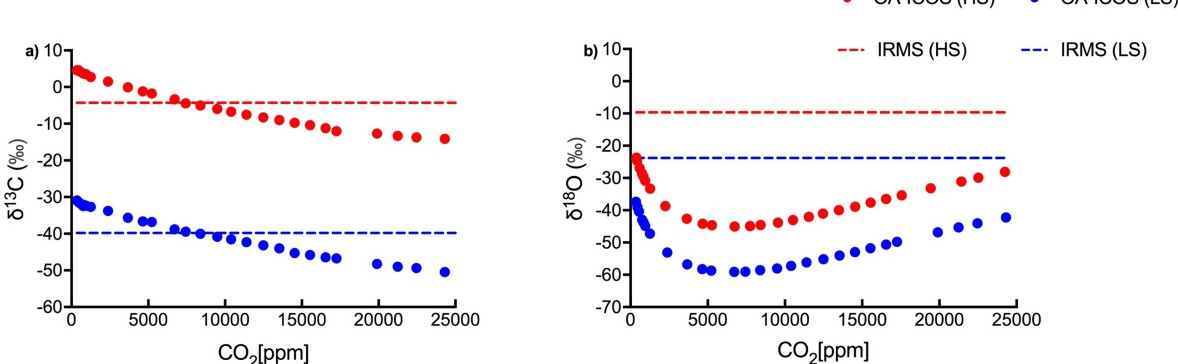

**Figure 4**: Variability observed in (a) $\delta^{13}C$ and (b) $\delta^{18}O$ measurements using OA-ICOS before calibration. $\delta^{13}C$ and $\delta^{18}O$ measured using OA-ICOS for Heavy Standard and Light Standard are shown as red and blue circles respectively. Actual $\delta^{13}C$ and $\delta^{18}O$ values reported after measuring by IRMS for heavy standard and light standard are shown as red and blue dashed lines respectively.

# Figure 5

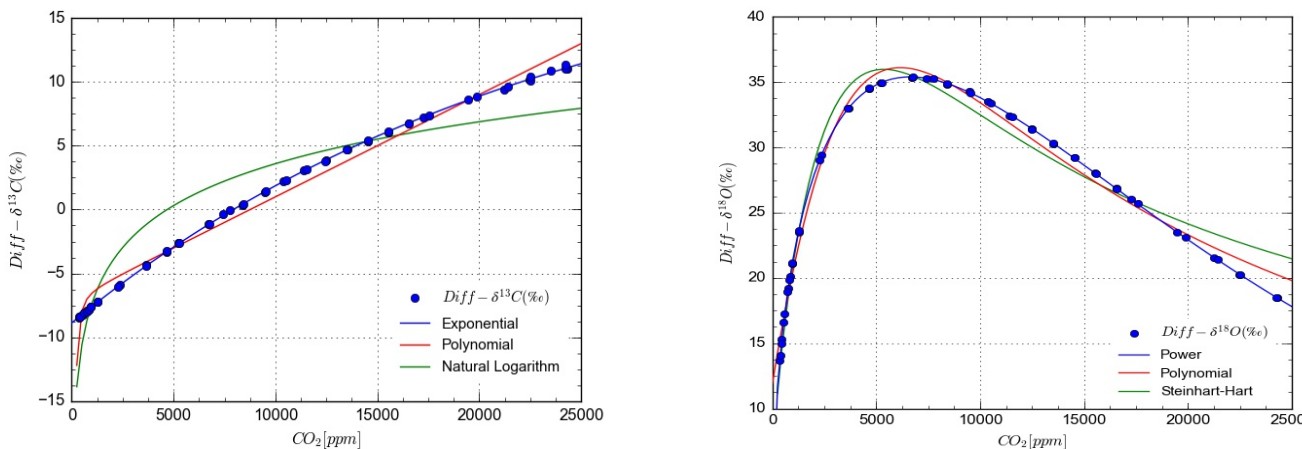

**Figure 5**: Mathematical models for concentration dependent drift in OA-ICOS measurements of stable isotopes of Carbon (a) and Oxygen (b) in $CO_2$ from IRMS measurements. Blue circles show Diff-$\delta^{13}$C (a) and Diff-$\delta^{18}$O (b) data points and lines represents different mathematical models fitted on the measured data.

# Figure 6

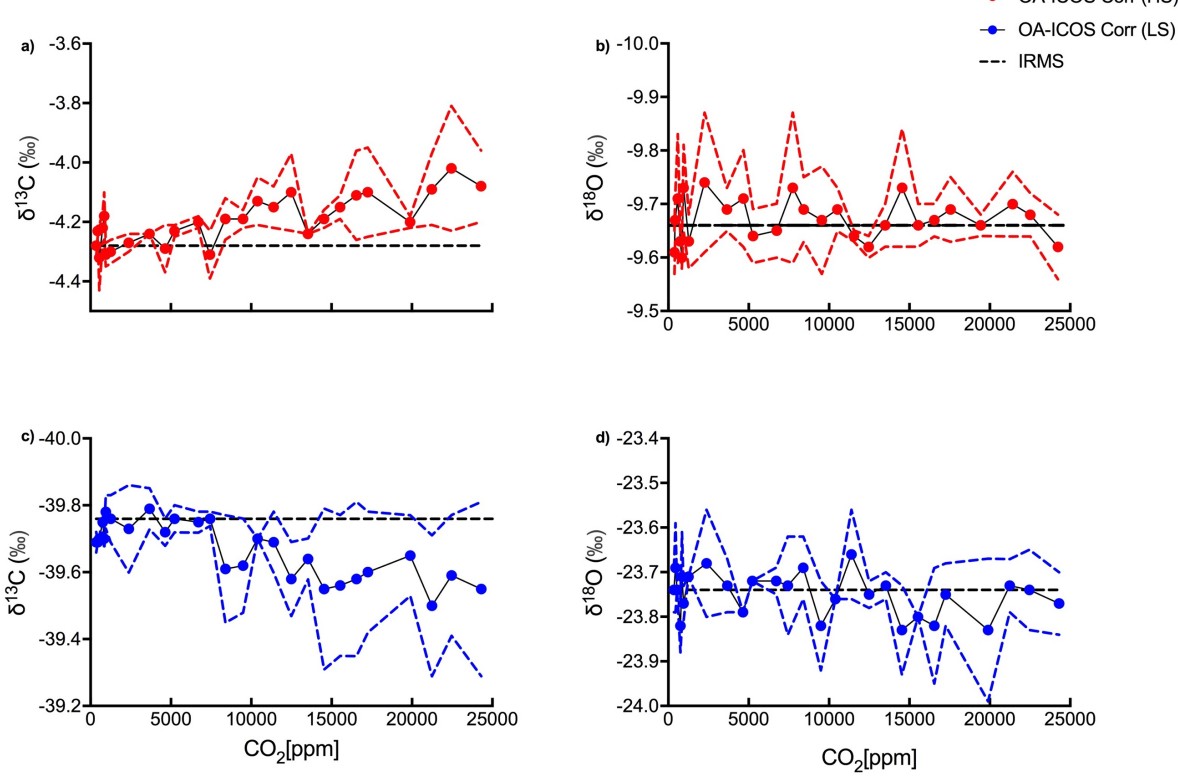

**Figure 6**: Corrected (a,c) $\delta^{13}C$ and (b,d) $\delta^{18}O$ measurements by OA-ICOS $CO_2$ Carbon isotope analyzer. $\delta^{13}C$ and $\delta^{18}O$ measured for Heavy Standard and Light Standard are shown as red and blue circles respectively. Actual $\delta^{13}C$ and $\delta^{18}O$ values reported after measuring by IRMS are shown as black dashed lines and 95% confidence intervals are shown as colored dashed lines respectively.

# Figure 7

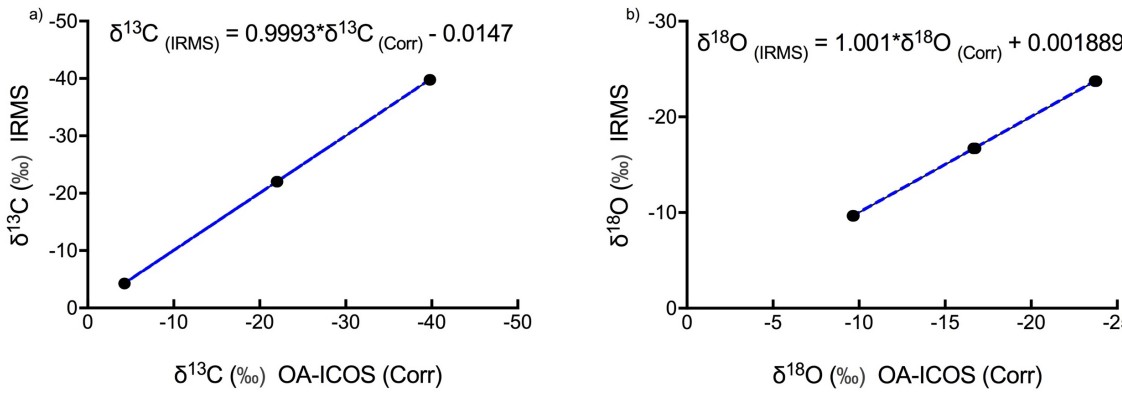

**Figure 7**: Three point Calibration lines for (a) $\delta^{13}C$ and (b) $\delta^{18}O$ measurements using OA-ICOS with 95% confidence interval.

# Figure 8

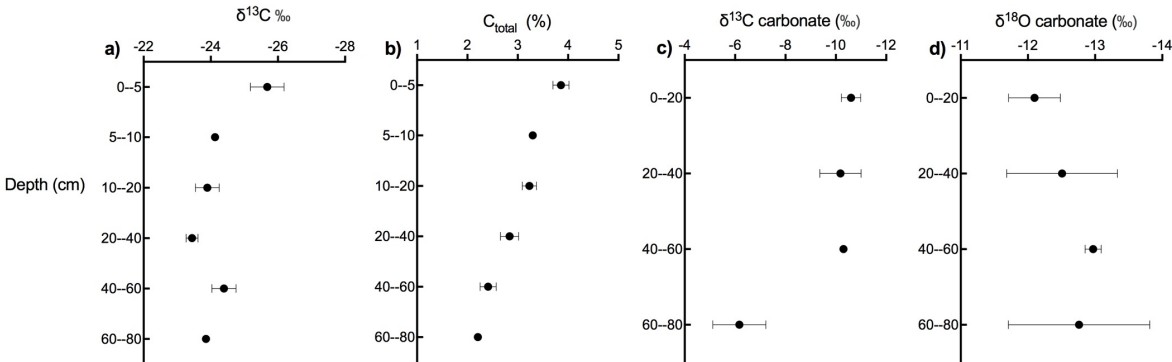

**Figure 8**: Depth profile of (a) δ¹³C, (b) Carbon content, (c) δ¹³C of soil carbonate and (d) δ¹⁸O of soil carbonate in calcareous soil.

# Figure 9

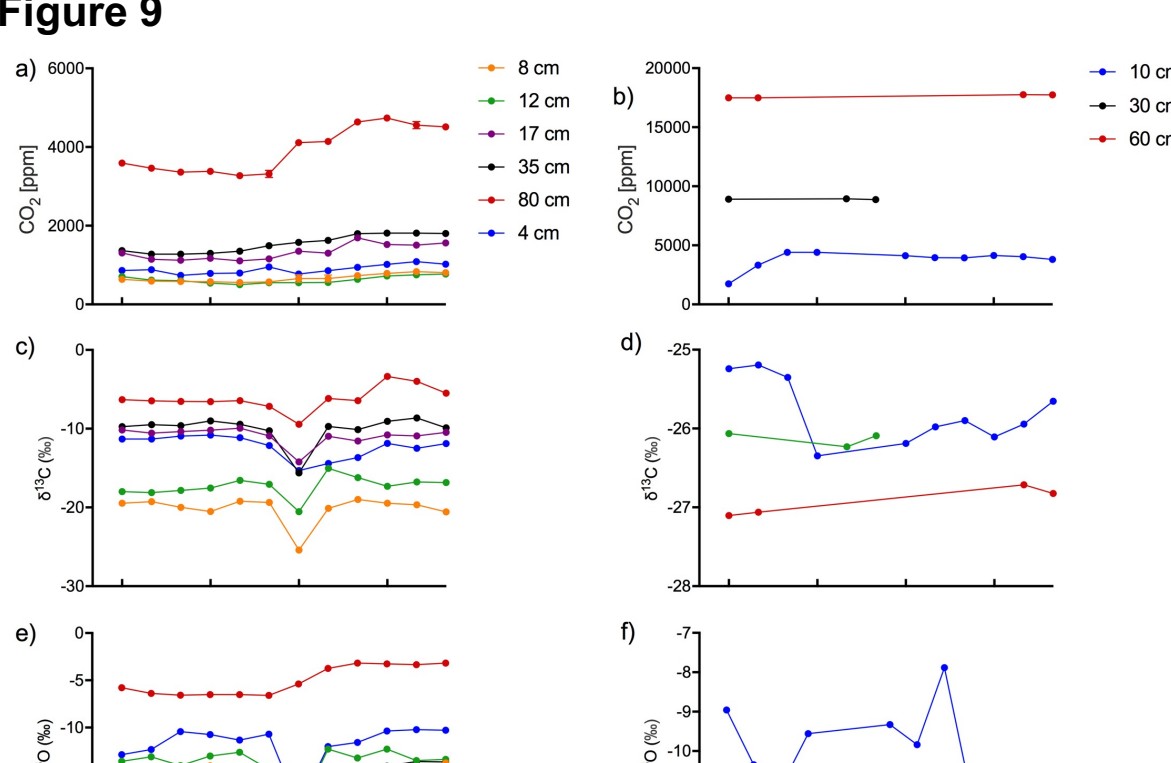

**Figure 9**: Time course of the evolution of soil gas $CO_2$ [ppm], $\delta^{13}C$ and $\delta^{18}O$ in calcareous (a,c,e) and acidic (b,d,f) soils. Data collected continuously over a 12 hour time frame for the calcareous soil and a 14 hour time window with intermittent data collection for the acidic soil.

# Figure 10

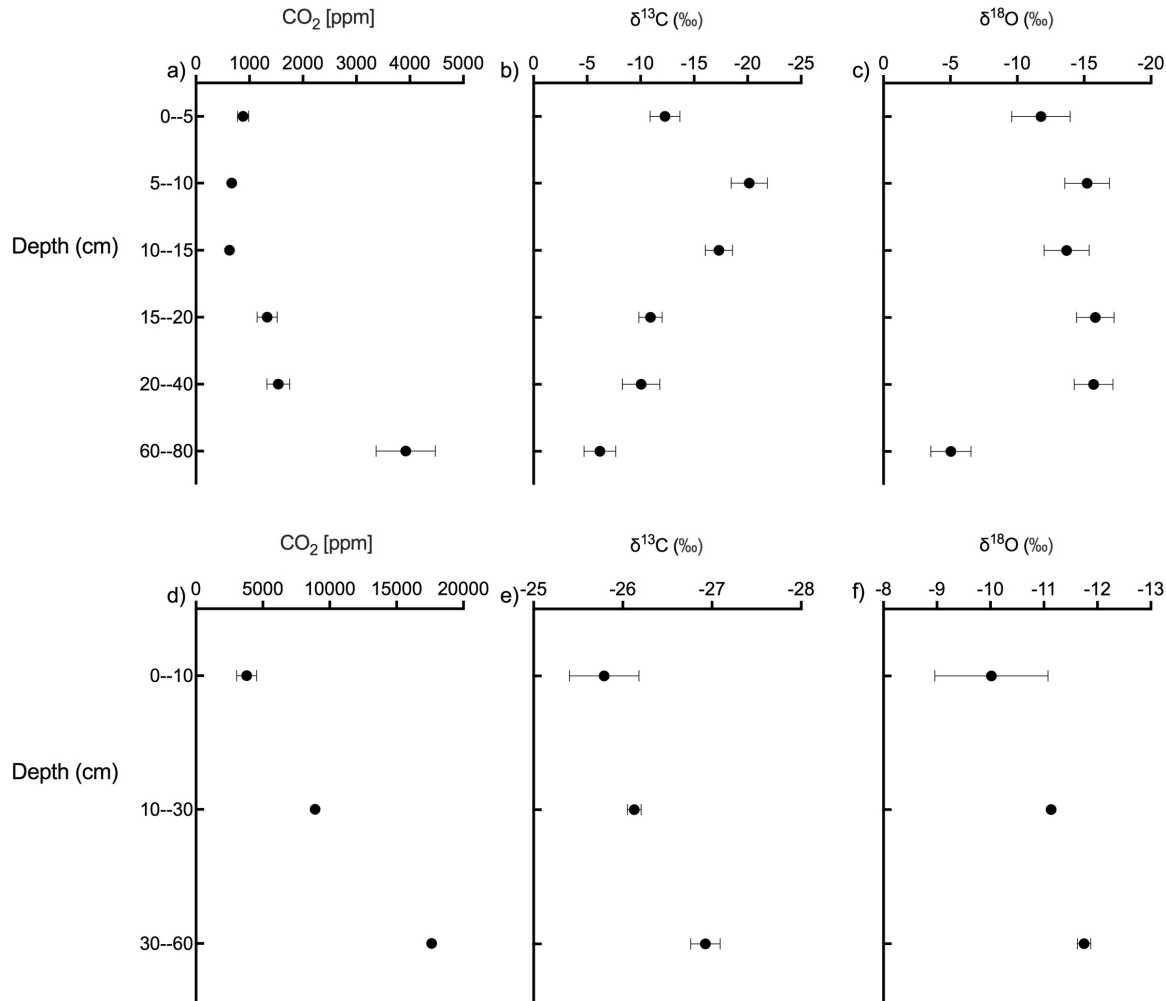

**Figure 10**: Daily average data of soil $CO_2$ [ppm], $\delta^{13}C$ and $\delta^{18}O$ in calcareous (a,b,c) and acidic (d,e,f) soils across soil depth profiles.