# Peer review of "Application of a laser-based spectrometer for continuous insitu measurements of stable isotopes of soil CO2 in calcareous and acidic soils"

_SOIL, 2018_

## Short Comment (SC1) · 11 Jun 2018

While the introduction section discussing soil CO2 measurement techniques does include some discussion of chamber and tower sampling, it should also include references that made use of soil gas wells to sample CO2 for d13C and d18O values, as that is more similar to the work the paper describes. I suggest you include these references, both of which are relevant to arid and semi-arid soils, as that is the focus of the manuscript under discussion:

Breecker, D., & Sharp, Z. D. (2008). A field and laboratory method for monitoring the concentration and isotopic composition of soil CO2. Rapid Communications in Mass

[Figure]

Spectrometry, 22(4), 449-454.

Oerter, E. J., & Amundson, R. (2016). Climate controls on spatial and temporal variations in the formation of pedogenic carbonate in the western Great Basin of North America. GSA Bulletin, 128(7-8), 1095-1104.

---

## Referee Comment (RC1) · Anonymous Referee #1 · 24 Jun 2018

This manuscript describes attempts to calibrate a laser-based absorption instrument for use in high-frequency measurements of 13C and 18O of CO2 in soil depth profiles, and provides very brief field data from two sites. This type of work is useful in that many investigators use new instruments, such as the LGR instrument described here, without sufficient validation. However, I have some major concerns about the calibration method which the authors can hopefully address in a revised manuscript. In addition, the paper would be much stronger if additional field data were presented, especially along with atmospheric measurements at the soil surface, which are needed to calculate the isotope composition of soil respired CO2 (as opposed to soil profile CO2). At present, it appears that <24 hours of field data are shown.

[Figure]

The authors develop non-linear calibration functions to account for the concentration dependence of isotope ratios but it is not clear to me how these functions might also vary in isotope space (i.e., as a dual function of isotope composition and concentration). For example, fig. 5 shows the correction functions for concentration dependence but does not show how/if these varied as a function of the isotope ratios of the standard gasses, which should all be shown on this figure. Furthermore, Fig 6 and 7 show serious deviations of calibrated vs. true values for both 13C and 18O between circa 2000 – 10000 ppm, of as much as 2 permil, even though those differences disappear at higher values. This deviation is unacceptable given the requirements of analyzing soil CO2, where differences of 2 permil may be highly significant from an ecological perspective. I note that CO2 concentrations < 10,000 ppm are commonplace in most soil profiles, especially in shallow horizons that typically dominate CO2 production, such that capacity for accurate and precise measurements in this lower concentration range is really critical. Even greater variability is shown in Fig 7, which appears to reach 4 per mil. This is not acceptable for natural abundance work.

I am skeptical as to the utility of the exponential calibration function given the unusual trends shown in Fig 4 that do not conform to an exponential curve (residual plots would clearly show violation). A spline fit may be needed and would likely avoid some of the problems mentioned above if it can be shown that the concentration response does not interact with the isotope ratio of the sample gas. Note the analogy from IRMS calibration, where separate calibration functions are typically used to correct for beam area effects and to normalize isotope ratios to the reference materials. A simple off-set is very seldom sufficient for adequate calibration. In Fig 7, we need to see how deviations from the 1:1 line vary as a function of isotope composition (what is driving the very large scatter for some observations?). This is another hint that isotope composition and CO2 mole fraction are interacting, implying that more complex calibration functions may be needed (maybe these were used and I am not understanding the method?). Also, it appears that independent standard gasses were not withheld to verify precision independently (i.e., standards that were not used to generate cal curves).

If this is not the case that should be clarified, and ideally at least one standard would be withheld for validation.

Diluting standard gasses with N2 is not good practice for calibrating measurements of CO2 in air samples due to the phenomenon of pressure broadening. The absence of O2 fundamentally changes the absorption properties of CO2 in an N2 matrix. There is abundant literature on this point, and it becomes especially important for isotope measurements.

Also, basic details about the soils investigated are missing that are necessary to interpret the measured values of 13C and 18O of CO2. For example, what are the carbonate concentrations and isotope ratios in the calcareous soil, and how do they vary with depth? What are the 13C values of SOM? This is a prerequisite for interpreting the soil profile CO2 values. Also, to calculate the isotope ratios of soil-respired CO2, we need measurements of the atmospheric boundary condition. See Davidson 1995 GCA, doi:10.1016/0016-7037(95)00143-3. Note that several recent papers neglected have reported 13C of CO2 from soil profiles using high temporal-resolution optical measurements, these should be discussed or at least mentioned.

I am skeptical as to the validity of the temperature tests employed. Note that we need to know the temperature of the analyte gas itself, which may be substantially different than the temperature of the water bath through which it circulates unless the residence time of the gas in the tubing and the heat transfer properties of the tubing allow for sufficiently rapid temperature equilibration, which may not completely occur if flow rates are high. For example, certain applications require heating of gasses at a sampling inlet to avoid condensation, yet the temperature of the gas at the point of the analysis may be substantially different (e.g. -4 – 40C) for some other optical gas analyzers, and should optimally be controlled within the analyzer cavity itself. Thus, unless the exact temperature of the gas at the point of measurement can be determined, I would not trust the results from the water bath experiment. Regardless, details of the analysis flow rate should be reported (and whether these rates were controlled during sample

analyses—MFC's are mentioned for standards only).

There are numerous issues with grammar, style, and errant capitalization throughout. The figures and tables have a strange mix of fonts (be consistent!) and the legends are compressed. Please follow standard procedures for presenting your MS (provide captions as text in the document, not as images). There is a significant typo in Table 1.

Finally, it should be noted that the useful temporal resolution of the measurements will never actually be 1hz as reported given the Allan variance results.

Was water vapor removed from the analyte gas, and if so, how?

―――――――――――――――――

---

## Referee Comment (RC2) · RTW Siegwolf (Referee) · 1 Aug 2018

**Comments to the manuscript    MS No.: soil-2018-9**

**Title:**    Application of a laser-based spectrometer for continuous insitu measurements of stable isotopes of soil CO2 in calcareous and acidic soils

**Authors**:  Jobin Joseph et al.

**A) Summary:**

This methodological paper describes the use of a laser spectrometer to measure the $CO_2$ concentration in the soil and its C and O isotope ratios from biotic (respiration) and abiotic sources. The authors suggest a new method to capture the soil-$CO_2$ with gas permeable, hydrophobic tubes, which were buried in different soil depths, to distinguish the different $CO_2$ origins. The released soil $CO_2$ was measured with an Off-Axis Cavity Output Spectrometer (OA-ICOS). As the measurements were done for a concentration range between 400 and 25´000 µmol/mol the authors described their calibration method of the laser instrument, an essential task when measuring such a wide concentration range. The authors then present and interpret their first results.

**B) General comments:**

This is a highly relevant contribution where the authors address an important issue regarding the measurements of $CO_2$ and its isotopic ratio over a large concentration range. The results demonstrate the importance of a careful calibration of the instruments, given the apparent nonlinearity between concentration and isotopic ratio. The manuscript is fairly well prepared, the objectives are clear although some passages in the text need clarification (see below). Besides, there are some methodological points and questions regarding the interpretation as indicated below that must be addressed before publication.

**General points:**

***Preparation of the calibration gases***: You mixed the gases in $N_2$. This will cause some shifts in your absorption spectra and will result in a shift of your isotopic values as it was shown in Bowling et al. (2003). Tuzon et al., (2008) address the calibration process in detail and it is recommended to consider this paper in this study. If the possibility is still given, it might be worth it produce new reference gases with synthetic $CO_2$ free air (20% oxygen and 80% nitrogen) then repeat the calibration of the instrument, compare the results and reassess the results. I am aware that this is an unusual request and almost too much to ask for but it would be worth it.

**How did you calibrate the gases**, via gas bench-IRMS or via cryo extraction and Dual Inlet IRMS? If you used the gas bench method how did you handle the problem with the septa of the vacutainers leading to a large scatter for the $^{18}O/^{16}O$ ratio, in case you used this method?

It would be worth to **insert subtitles** in chapter 3: e.g.
3.1 Instrument calibration and correction (after Line 192)
3.2 Variation in soil $CO_2$ concentration and its C and O isotope values (after line 241)

**Specific comments**

**Line 140**: PTFE or Swagelok filter? Clarify

**Line Lines 141-142**: what kind of a filter is this to prevent moisture from getting into the device? What device do you mean? Normally moisture isn´t captured with a filter but much rather with a water trap. But usually commercially available gas is very dry making a water trap dispensable.

**Lines 145-146**: If you intend to produce a gas with a temperature range from minus! -20°C to +40°C a **water** bath is certainly not the right choice. Please clarify. Either you used a different cooling liquid or you never went below 0°C

**Line 156**: Please indicate the concentration steps for the calibration.

**Line 187**: How was the pressure regulated in this closed loop? For a proper operation of the laser instrument, the pressure in the cavity cell must be as constant as possible, since only slightest changes in pressure can mimic a change in concentration of all gas species.

**Line 204**: To prevent misunderstandings it is better to write D-$\delta$ or Diff-$\delta$ instead of $\Delta\delta$, since $\Delta$ is used for discrimination (fractionation) in the isotope literature.

**Line 206**: rewrite "… *The mathematical model with the most fitting to*…" write "…*the mathematical model with the best fir for …*"

**Line 211**: replace "… *most fitting model …*" with "… *best fit …*"

**Line 221**: replace "… *better…*" with " … *the needed…*"

**Lines 223 – 231:** A native English speaking person should reassess these lines.

**Lines 226- 227**: It would be more correct to say: " We assume that these **deviations** were instrument specific and the **fitting parameters** have to be adjusted for every single device.

**Lines 243-245**. I can´t see that for the top 4 to 12cm. Clarify please.

**Line 246**: …relative to what? Soil $\delta^{13}CO_2$ was only slightly enriched, according to Fig. 8

**Lines 242-272**: For this whole paragraph it would be worth to read the paper of Cerling, 1984, and Bowen, 2004 (see recommended literature).

**Line 250**: No specific pattern…Actually the pattern for $\delta^{18}O$ is quite similar to that of the $\delta^{13}C$, except for this sharp decline at around 2:00, (which is less visible for the $\delta^{13}C$ time course) . The authors should comment that, what could be the cause?

**Line 254**: It would be highly beneficial for this statement if you had the $\delta$ values of the soil organic matter for the respective soil depths.

**Line 264**: It would be more accurate to say: "…*is assumed to be the dominating source of soil CO$_2$…*"

**Lines 269-272**: Are you sure that the $\delta^{18}O$ values of the soil $CO_2$ are referred to VSMOW? It looks more like VPDB. Please check that! Then, compared to the $\delta^{18}O$ values close to the soil surface $CO_2$ the $\delta^{18}O$ values in -80 cm depth are surprisingly high relative to the topsoil. Soil surface water is more prone to be enriched, due to soil surface evaporation processes, than water close to ground water. The authors should comment on that.

**Lines 281-283:** Here it would be valuable to have more information on the soil structure. Isn´t the acidic soil less compact and dense than the calcareous soil and therefore the diffusivity would be higher in the acidic soil. Its higher $CO_2$ concentration could as well be a result of a higher microbial activity due to its higher organic content. It would be interesting to see soil respiration data for these soils. Maybe the authors can comment on that

**Lines 285-287:** Again are these $\delta^{18}O$ values really referring to the VSMOW scale? Then somehow your calculation between the $\delta^{18}O$ of the soil water and that of the $CO_2$ is strange. If you add 41‰ (oxygen fractionation between water and $CO_2$) to -10‰ ($\delta^{18}O$ of the soil water) that would result in ca. 31‰, but you indicate -10‰. Please clarify.

**Conclusion**: The first 8 lines are more a summary than a conclusion. Focus on the main outcome of your study, which is the non-linear response of the $\delta$-values versus $CO_2$ concentration. This is a strong demonstration for how **essential** a careful concentration vs. Isotope ratio calibration is especially when the system is used for such a wide concentration range. Then it would be interesting if your tube-soil-$CO_2$-capture method is reliable and highlight the advantages and disadvantages versus other methods. You practically ignored this method in the discussion. It would be interesting to know more about your experience with it. In that light what do you conclude from your first results?

**Figures:**

In all Figures, where you plot $\delta^{18}O$ values, check whether you used the VSMOW or VPDB scale.

Fig. 1: the expression "water bath" is misleading better to use an expression like "gas thermostat system" or something alike. Clarify whether you used PTFE (brand, type, producer etc.) or Swagelok filters.

Fig 5 and Fig 6: it would be better to use D-$\delta$ or Diff-$\delta$ instead of $\Delta\delta$

Fig. 8: Indicate in the figure legend that this is a "… **Time course** of the evolution of …" with the specific time resolution.

Fig. 9: Indicate in the figure legend that you display "…**Daily? averages** of $CO_2$ concentration and isotope values in depth profiles…"

**Literature worth reading and potentially including:**

*About laser spectrometer calibration and application*

**Bowling D**. et al., (2003) Tunable diode laser absorption spectroscopy for stable isotope studies of ecosystem–atmosphere $CO_2$ exchange  Agricultural and Forest Meteorology, 118, 1-19.

**Tuzson B**. et al., (2008) High precision and continuous field measurements of d$^{13}$C and d$^{18}$O in carbon dioxide with a cryogen-free QCLAS. Applied Physics B – Lasers and Optics; 92, 451–458, DOI: 10.1007/s00340-008-3085-4

Nelson D. et al., (2008) New method for isotopic ratio measurements of atmospheric carbon dioxide using a 4.3 μm pulsed quantum cascade laser  Applied; Physics B – Lasers and Optics, 90, 301–309 (2008); DOI: 10.1007/s00340-007-2894-1

Sturm et al., (2012) Eddy covariance measurements of $CO_2$ isotopologues with a quantum cascade laser absorption spectrometer. Agricultural and Forest Meteorology 152, 73–82.

**Wehr R**., S.R. Saleska  (2015) An improved isotopic method for partitioning net ecosystem–atmosphere $CO_2$ exchange. Agricultural and Forest Meteorology 214-215 (2015) 515–531.

Wehr R. et al., (2016) Seasonality of temperate forest photosynthesis and daytime respiration. NATURE | VOL 534 | 30 jUNE 2016

**About isotopes in soil CO2**

**Kato** et al., (2004) Seasonal variation of the oxygen isotopic ratio of atmospheric carbon dioxide in a temperate forest, Japan. GLOBAL BIOGEOCHEMICAL CYCLES, VOL. 18, GB2020, doi:10.1029/2003GB002173

**Cerling** TE (1984) The stable isotopic composition of modern soil carbonate and its relationship to climate. Earth and Planetary Science Letters, 71 (1984) 229-240

Bowen  G. (2004) An integrated model for soil organic carbon and $CO_2$: Implications for paleosol carbonate $pCO_2$ paleobarometry. GLOBAL BIOGEOCHEMICAL CYCLES, VOL. 18, GB1026, doi:10.1029/2003GB002117

---

## Author Comment (AC1) · 30 Sep 2018

Reviewer 1# Remarks to the Author

This manuscript describes attempts to calibrate a laser-based absorption instrument for use in high-frequency measurements of 13C and 18O of CO2 in soil depth profiles, and provides very brief field data from two sites. This type of work is useful in that many investigators use new instruments, such as the LGR instrument described here, without sufficient validation. However, I have some major concerns about the calibration method which the authors can hopefully address in a revised manuscript. In addition, the paper would be much stronger if additional field data were presented, especially

along with atmospheric measurements at the soil surface, which are needed to calculate the isotope composition of soil respired CO2 (as opposed to soil profile CO2). At present, it appears that <24 hours of field data are shown.

1. The authors develop non-linear calibration functions to account for the concentration dependence of isotope ratios but it is not clear to me how these functions might also vary in isotope space (i.e., as a dual function of isotope composition and concentration). For example, fig. 5 shows the correction functions for concentration dependence but does not show how/if these varied as a function of the isotope ratios of the standard gasses, which should all be shown on this figure. Furthermore, Fig 6 and 7 show serious deviations of calibrated vs. true values for both 13C and 18O between circa 2000 – 10000 ppm, of as much as 2 permil, even though those differences disappear at higher values. This deviation is unacceptable given the requirements of analyzing soil CO2, where differences of 2 permil may be highly significant from an ecological perspective. I note that CO2 concentrations < 10,000 ppm are commonplace in most soil profiles, especially in shallow horizons that typically dominate CO2 production, such that capacity for accurate and precise measurements in this lower concentration range is really critical. Even greater variability is shown in Fig 7, which appears to reach 4 per mil. This is not acceptable for natural abundance work.

Response: As per the suggestions made by both of the referees, we have conducted another round of calibration by diluting CO2 gas using synthetic air instead of N2. Diluting the CO2 standard gas with N2 resulted in a standard deviation of 8.1(‰ for $\delta$13C values and 4.7 (‰ for $\delta$18O values respectively. Diluting CO2 standard gases with synthetic air resulted in a standard deviation of 6.44(‰ and 6.818(‰ for $\delta$13C and $\delta$18O respectively (see Fig.1a-b). With our new calibration curves (see Fig.1 c-d, &Table.1,2), we are able to bring down standard deviation to 0.08(‰ for $\delta$13C and 0.04(‰ for $\delta$18O (see Fig.2a-b (residual distribution), Fig.3a-d (Corrected $\delta$13C and $\delta$18O values). By introducing the new calibration correction (see.Fig.3) the values are very close to the target value across the whole concentration range and hence we are

confident that the system is suitable for ecosystem studies based on measuring subtle changes in isotopic signature of CO2 across plant soil atmosphere continuum.

2. Also, basic details about the soils investigated are missing that are necessary to interpret the measured values of 13C and 18O of CO2. For example, what are the carbonate concentrations and isotope ratios in the calcareous soil, and how do they vary with depth? What are the 13C values of SOM? This is a prerequisite for interpreting the soil profile CO2 values. Also, to calculate the isotope ratios of soil-respired CO2, we need measurements of the atmospheric boundary condition. See Davidson 1995 GCA, doi:10.1016/0016-7037(95)00143-3. Note that several recent papers neglected have reported 13C of CO2 from soil profiles using high temporal-resolution optical measurements, these should be discussed or at least mentioned.

Response: We have measured soil samples for bulk $\delta$13C, bicarbonate $\delta$13C & $\delta$18O values and also % of carbon content in the soil across a depth profile of (0-80 cm) for the calcareous soil (See Fig.4a-c and Fig.5). We observed an increase in $\delta$13C values (of bulk soil and carbonate) in deeper soil layers (See Fig.4 a,c). This fits to our assumption of an increased contribution of bicarbonate derived $\delta$13C enriched CO2 in deeper soil layers. Our aim is to establish a method which enables continuous online measurement of soil gas $\delta$13C & $\delta$18O values at different soil depths and hence calculating the isotope ratios of soil-respired CO2 is not done in this manuscript. This would be beyond the scope of a calibration focused paper – we however show the importance to properly calibrate laser based systems to obtain valid measurements of d13C and d18O of soil CO2 which is a prerequisite for assessing the rate and isotopic composition of soil respiration.

3. I am skeptical as to the validity of the temperature tests employed. Note that we need to know the temperature of the analyte gas itself, which may be substantially different than the temperature of the water bath through which it circulates unless the residence time of the gas in the tubing and the heat transfer properties of the tubing allow for sufficiently rapid temperature equilibration, which may not completely occur if flow rates

are high. For example, certain applications require heating of gasses at a sampling inlet to avoid condensation, yet the temperature of the gas at the point of the analysis may be substantially different (e.g. -4 – 40C) for some other optical gas analyzers, and should optimally be controlled within the analyzer cavity itself. Thus, unless the exact temperature of the gas at the point of measurement can be determined, I would not trust the results from the water bath experiment. Regardless, details of the analysis flow rate should be reported (and whether these rates were controlled during sample analysesâAËŸTMFC's are mentioned for standards only).

Response: The laser spectrometer was able to maintain the temperature inside the optical cavity quite stable at 46.61°C irrespective of the fluctuations in the gas temperature (See Figure attached below). It is clear that the temperature maintained in the water bath will not get directly reflected in the sample gas due to multiple reasons including diffusion barrier of the PTFE tubing and higher flow rates, never the less, there will be an increment or decrement in the gas temperature. The aim is to show that the system is also stable in field conditions where temperature fluctuation is happening. The system is running in a closed loop meaning there is enough time for the gas for heat exchange. We have adjusted the part where test for equipment stability under fluctuating temperature is done in the modified manuscript.

4. There are numerous issues with grammar, style, and errant capitalization throughout. The figures and tables have a strange mix of fonts (be consistent!) and the legends are compressed. Please follow standard procedures for presenting your MS (provide captions as text in the document, not as images). There is a significant typo in Table 1.

Response: This is addressed and rectified in the revised manuscript. We will also let a native speaker do the final editing of the manuscript.

5. Finally, it should be noted that the useful temporal resolution of the measurements will never actually be 1hz as reported given the Allan variance results.

Response: Not exactly clear what the reviewer meant by "it should be noted that the

useful temporal resolution of the measurements will never actually be 1hz as reported given the Allan variance results". It is always useful to get a better temporal resolution which can be used for identifying short term dynamics of CO2 efflux (e.g., diurnal pattern of soil CO2 efflux). Meaning more data points are available for taking an average across a time frame which is best for noise correction by using Allan deviation curves.

6. Was water vapor removed from the analyte gas, and if so, how? Response: Yes, water vapor was removed using drierite desiccant cartridges. We will add this information to the revised manuscript.

Figure.1: Deviation of measured $\delta$13C and $\delta$18O of CO2 (a,b) when diluted using synthetic air. (c-d) shows diff- $\delta$13C, diff-$\delta$18O values across a concentration gradient. Red and Blue dots shows measured $\delta$13C & $\delta$18O values of two different gases with distinct isotopic signatures, red and blue dashed lines represents absolute $\delta$13C & $\delta$18O values of the respective gases. Black line denotes model fit for diff- $\delta$13C, diff-$\delta$18O values across changing CO2 concentration (300 – 25000 ppm).

Figure 2: Residual distribution of modeled data for diff- $\delta$13C, diff-$\delta$18O values across changing CO2 concentration (300 – 25000 ppm).

Figure 3: Corrected $\delta$13C & $\delta$18O values of two different standard gases measured after correcting for concentration dependent drift.

Figure.4: Bulk $\delta$13C (a), bicarbonate $\delta$13C (c) and % of carbon (b) in soil across a depth profile of (0-80 cm).

Figure.5: bicarbonate $\delta$18O in soil across a depth profile of (0-80 cm).

Please also note the supplement to this comment:
https://www.soil-discuss.net/soil-2018-9/soil-2018-9-AC1-supplement.pdf
* * *
[Figure]

Fig.1

a)

b)

c)

d)

Gas A
Gas B
Absolute
Absolute
Model

$CO_2$[ppm]

$CO_2$[ppm]

δ$^{13}$C (‰)

δ$^{18}$O (‰)

Diff-δ$^{13}$C (‰)

Diff-δ$^{18}$O (‰)

**Fig. 1.**

[Figure]

[Figure]

**Fig. 2.**

[Figure]

Fig.3

a)

b)

Gas A
Gas B
--- Absolute

c)

d)

CO₂[ppm]

CO₂[ppm]

**Fig. 3.**

[Figure]

Fig. 4.

[Figure]

[Figure]

[Figure]

Fig.5

**Fig. 5.**

[Figure]

Fig. 6.

**Supplement:**

This manuscript describes attempts to calibrate a laser-based absorption instrument for use in high-frequency measurements of 13C and 18O of CO2 in soil depth profiles, and provides very brief field data from two sites. This type of work is useful in that many investigators use new instruments, such as the LGR instrument described here, without sufficient validation. However, I have some major concerns about the calibration method which the authors can hopefully address in a revised manuscript. In addition, the paper would be much stronger if additional field data were presented, especially along with atmospheric measurements at the soil surface, which are needed to calculate the isotope composition of soil respired CO2 (as opposed to soil profile CO2). At present, it appears that <24 hours of field data are shown.

1.  The authors develop non-linear calibration functions to account for the concentration dependence of isotope ratios but it is not clear to me how these functions might also vary in isotope space (i.e., as a dual function of isotope composition and concentration). For example, fig. 5 shows the correction functions for concentration dependence but does not show how/if these varied as a function of the isotope ratios of the standard gasses, which should all be shown on this figure. Furthermore, Fig 6 and 7 show serious deviations of calibrated vs. true values for both 13C and 18O between circa 2000 – 10000 ppm, of as much as 2 permil, even though those differences disappear at higher values. This deviation is unacceptable given the requirements of analyzing soil CO2, where differences of 2 permil may be highly significant from an ecological perspective. I note that CO2 concentrations < 10,000 ppm are commonplace in most soil profiles, especially in shallow horizons that typically dominate CO2 production, such that capacity for accurate and precise measurements in this lower concentration range is really critical. Even greater variability is shown in Fig 7, which appears to reach 4 per mil. This is not acceptable for natural abundance work.

Response: As per the suggestions made by both of the referees, we have conducted another round of calibration by diluting CO2 gas using synthetic air instead of N2. Diluting the CO2 standard gas with N2 resulted in a standard deviation of 8.1(‰) for $\delta^{13}$C values and 4.7 (‰) for $\delta^{18}$O values respectively. Diluting CO2 standard gases with synthetic air resulted in a standard deviation of 6.44(‰) and 6.818(‰) for $\delta^{13}$C and $\delta^{18}$O respectively (see Fig.1a-b). With our new calibration curves (see Fig.1 c-d, &Table.1,2), we are able to bring down standard deviation to 0.08(‰) for $\delta^{13}$C and 0.04(‰) for $\delta^{18}$O (see Fig.2a-b (residual distribution), Fig.3a-d (Corrected $\delta^{13}$C and $\delta^{18}$O values). By introducing the new calibration correction (see.Fig.3) the values are very close to the target value across the whole concentration range and hence we are confident that the system is suitable for ecosystem

studies based on measuring subtle changes in isotopic signature of CO2 across plant soil atmosphere continuum.

[Figure]

Figure.1: Deviation of measured $\delta^{13}C$ and $\delta^{18}O$ of CO2 (a,b) when diluted using synthetic air. (c-d) shows diff- $\delta^{13}C$, diff-$\delta^{18}O$ values across a concentration gradient. Red and Blue dots shows measured $\delta^{13}C$ & $\delta^{18}O$ values of two different gases with distinct isotopic signatures, red and blue dashed lines represents absolute $\delta^{13}C$ & $\delta^{18}O$ values of the respective gases. Black line denotes model fit for diff- $\delta^{13}C$, diff-$\delta^{18}O$ values across changing CO2 concentration (300 – 25000 ppm).

[Figure]

[Figure]

Figure 2: Residual distribution of modeled data for diff- $\delta^{13}C$, diff-$\delta^{18}O$ values across changing CO2 concentration (300 – 25000 ppm).

[Figure]

Figure 3: Corrected $\delta^{13}C$ & $\delta^{18}O$ values of two different standard gases measured after correcting for concentration dependent drift.

2. Also, basic details about the soils investigated are missing that are necessary to interpret the measured values of 13C and 18O of CO2. For example, what are the carbonate concentrations and isotope ratios in the calcareous soil, and how do they vary with depth? What are the 13C values of SOM? This is a prerequisite for interpreting the soil profile CO2 values. Also, to calculate the isotope ratios of soil-respired CO2, we need measurements of the atmospheric boundary condition. See Davidson 1995 GCA, doi:10.1016/0016-7037(95)00143-3. Note that several recent papers neglected have reported 13C of CO2 from soil profiles using high temporal-resolution optical measurements, these should be discussed or at least mentioned.

[Figure]

Figure.4: Bulk $\delta^{13}C$ (a), bicarbonate $\delta^{13}C$ (c) and % of carbon (b) in soil across a depth profile of (0-80 cm).

[Figure]

Figure.5: bicarbonate $\delta^{18}O$ in soil across a depth profile of (0-80 cm).

Response: We have measured soil samples for bulk $\delta^{13}C$, bicarbonate $\delta^{13}C$ & $\delta^{18}O$ values and also % of carbon content in the soil across a depth profile of (0-80 cm) for the calcareous soil (See Fig.4a-c and Fig.5). We observed an increase in $\delta^{13}C$ values (of bulk soil and carbonate) in deeper soil layers (See Fig.4 a,c). This fits to our assumption of an increased contribution of

bicarbonate derived $\delta^{13}C$ enriched CO2 in deeper soil layers.. Our aim is to establish a method which enables continuous online measurement of soil gas $\delta^{13}C$ & $\delta^{18}O$ values at different soil depths and hence calculating the isotope ratios of soil-respired CO2 is not done in this manuscript. This would be beyond the scope of a calibration focused paper – we however show the importance to properly calibrate laser based systems to obtain valid measurements of d13C and d18O of soil CO2 which is a prerequisite for assessing the rate and isotopic composition of soil respiration.

I am skeptical as to the validity of the temperature tests employed. Note that we need to know the temperature of the analyte gas itself, which may be substantially different than the temperature of the water bath through which it circulates unless the residence time of the gas in the tubing and the heat transfer properties of the tubing allow for sufficiently rapid temperature equilibration, which may not completely occur if flow rates are high. For example, certain applications require heating of gasses at a sampling inlet to avoid condensation, yet the temperature of the gas at the point of the analysis may be substantially different (e.g. -4 – 40C) for some other optical gas analyzers, and should optimally be controlled within the analyzer cavity itself. Thus, unless the exact temperature of the gas at the point of measurement can be determined, I would not trust the results from the water bath experiment. Regardless, details of the analysis flow rate should be reported (and whether these rates were controlled during sample analysesâ˘TMFC's are mentioned for standards only).

Response: The laser spectrometer was able to maintain the temperature inside the optical cavity quite stable at 46.61°C irrespective of the fluctuations in the gas temperature (See Figure attached below). It is clear that the temperature maintained in the water bath will not get directly reflected in the sample gas due to multiple reasons including diffusion barrier of the PTFE tubing and higher flow rates, never the less, there will be an increment or decrement in the gas temperature. The aim is to show that the system is also stable in field conditions where temperature fluctuation is happening. The system is running in a closed loop meaning there is enough time for the gas for heat exchange. We have adjusted the part where test for equipment stability under fluctuating temperature is done in the modified manuscript.

[Figure]

3.  There are numerous issues with grammar, style, and errant capitalization throughout. The figures and tables have a strange mix of fonts (be consistent!) and the legends are compressed. Please follow standard procedures for presenting your MS (provide captions as text in the document, not as images). There is a significant typo in Table 1.

Response: This is addressed and rectified in the revised manuscript. We will also let a native speaker do the final editing of the manuscript.

4.  Finally, it should be noted that the useful temporal resolution of the measurements will never actually be 1hz as reported given the Allan variance results.

Response: Not exactly clear what the reviewer meant by "it should be noted that the useful temporal resolution of the measurements will never actually be 1hz as reported given the Allan variance results". It is always useful to get a better temporal resolution which can be used for identifying short term dynamics of CO2 efflux (e.g., diurnal pattern of soil CO2 efflux). Meaning more data points are available for taking an average across a time frame which is best for noise correction by using Allan deviation curves.

5.  Was water vapor removed from the analyte gas, and if so, how?

Response: Yes, water vapor was removed using drierite desiccant cartridges. We will add this information to the revised manuscript.

Table.1

| Equation | y=a*(b-exp(-c*x)) | | | | |
|---|---|---|---|---|---|
| Standard Error | 0.07468171 | | | | |
| Correlation Coeff.(r) | 0.999941 | | | | |
| Coeff.of Determination (r^2) | 0.99988246 | | | | |
| DOF | 54 | | | | |
| AICC | -294.6349 | | | | |
| Parameters | | | | | |
| Value | Std | Err | Range | (95% | confidence) |
| a | 31.007446 | 0.214984 | 30.576428 | to | 31.438463 |
| b | 0.713759 | 0.002376 | 0.708995 | to | 0.718522 |
| c | 0.000043 | 0 | 0.000042 | to | 0.000043 |
| | | | | | |
| Covariance Matrix | | | | | |
| | a | b | c | | |
| a | 8.286768 | 0.088333 | -0.000018 | | |
| b | 0.088333 | 0.001012 | 0 | | |
| c | -0.000018 | 0 | 0 | | |

Table.2

| Equation | y=a*(b^x)*(x^c) | | | | |
|---|---|---|---|---|---|
| Standard Error | 0.04365503 | | | | |
| Correlation Coeff.(r) | 0.999981 | | | | |
| Coeff.of Determination (r^2) | 0.99996128 | | | | |
| DOF | 51 | | | | |
| AICC | -337.04644 | | | | |
| Parameters | | | | | |
| | Value | StdErr | Err | Range (95% confidence) | |
| a | 0.851623 | 0.003025 | 0.84555 | to | 0.857697 |
| b | 0.999928 | 0 | 0.999928 | to | 0.999928 |
| c | 0.477819 | 0.000472 | 0.476871 | to | 0.478767 |
| | | | | | |
| Covariance Matrix | | | | | |
| | a | b | c | | |
| a | 0.004803 | 0 | -0.000745 | | |
| b | 0 | 0 | 0 | | |
| c | -0.000745 | 0 | 0.000117 | | |

---

## Author Comment (AC2) · 30 Sep 2018

Reviewer 2# Remarks to the Author

1. Preparation of the calibration gases: You mixed the gases in N2. This will cause some shifts in your absorption spectra and will result in a shift of your isotopic values as it was shown in Bowling et al. (2003). Tuzon et al., (2008) address the calibration process in detail and it is recommended to consider this paper in this study. If the possibility is still given, it might be worth it produce new reference gases with synthetic CO2 free air (20% oxygen and 80% nitrogen) then repeat the calibration of the instrument, compare the results and reassess the results. I am aware that this is an unusual

request and almost too much to ask for but it would be worth it.

Response: As per the suggestions made by both of the referees, we have conducted another round of calibration by diluting the $CO_2$ gas using synthetic air (20% oxygen and 80% nitrogen) instead of N2. Diluting the $CO_2$ standard gas with N2 resulted in a standard deviation of 8.1(‰ for $\delta13C$ values and 4.7 (‰ for $\delta18O$ values respectively. Diluting $CO_2$ standard gases with synthetic air resulted in a standard deviation of 6.44(‰ and 6.818(‰ for $\delta13C$ and $\delta18O$ respectively (see Fig.1a-b). With our new calibration curves (see Fig.1 c-d, &Table.1,2), we are able to bring down standard deviation to 0.08(‰ for $\delta13C$ and 0.04(‰ for $\delta18O$ (see Fig.2a-b (residual distribution), Fig.3a-d (Corrected $\delta13C$ and $\delta18O$ values). We will restructure and include new calibration system in the revised version of the manuscript.

2. How did you calibrate the gases, via gas bench-IRMS or via cryo extraction and Dual Inlet IRMS? If you used the gas bench method how did you handle the problem with the septa of the vacutainers leading to a large scatter for the 18O/16O ratio, in case you used this method?

Response: The $\delta13C$ and $\delta18O$ values of our inhouse calibration gas standards were measured via cryo extraction and Dual Inlet IRMS. This will be included in a revised version of the manuscript.

3. It would be worth to insert subtitles in chapter 3: e.g. 3.1 Instrument calibration and correction (after Line 192) 3.2 Variation in soil $CO_2$ concentration and its C and O isotope values (after line 241).

Response: Subtitles will be added in the modified manuscript.

Specific comments

4. Line 140: PTFE or Swagelok filter? Clarify

Response: Swagelok filter (Stainless Steel In-Line Particulate Filter, 6 mm Swagelok Tube Fitting, 15 Micron Pore Size); this information will be added to the revised version.

5. Lines 141-142: what kind of a filter is this to prevent moisture from getting into the device? What device do you mean? Normally moisture isn't captured with a filter but much rather with a water trap. But usually commercially available gas is very dry making a water trap dispensable.

Response: The filter is a particulate matter filter and not a moisture filter. It can hold very little amount of liquid water and not water vapor. This will be rectified in the revised manuscript.

6. Lines 145-146: If you intend to produce a gas with a temperature range from minus! -20°C to +40°C a water bath is certainly not the right choice. Please clarify. Either you used a different cooling liquid or you never went below 0°C

Response: The reviewer is right, it needs further clarification. We have used a water bath to increase the temperature to higher values than the room temperature. To reduce the temperature below, we immersed gas tubes in liquid Nitrogen kept in an isotherm flask. This information will be included in the revised manuscript.

7. Line 156: Please indicate the concentration steps for the calibration.

Response: We have used 27 concentration points across the range (300-25000 ppm). For more details see table.3.

8. Line 187: How was the pressure regulated in this closed loop? For a proper operation of the laser instrument, the pressure in the cavity cell must be as constant as possible, since only slightest changes in pressure can mimic a change in concentration of all gas species.

Response: We did not encounter any pressure differences while maintaining a closed loop system. We have cavity pressure data monitored (see Figure.7). We will include this information in the revised version of the manuscript.

9. Line 204: To prevent misunderstandings it is better to write D-$\delta$ or Diff-$\delta$ instead of $\Delta\delta$, since $\Delta$ is used for discrimination (fractionation) in the isotope literature.

Response: We agree with the reviewer. It will be changed in the revised manuscript.

10. Line 206: rewrite "... The mathematical model with the most fitting to..." write "...the mathematical model with the best fir for ..."

Response: Corrected in the revised manuscript.

11. Line 211: replace "... most fitting model ..." with "... best fit ..." Response: Corrected in the revised manuscript.

12. Line 221: replace "... better..." with " ... the needed..." Response: Corrected in the revised manuscript.

13. Lines 223 – 231: A native English-speaking person should reassess these lines.

Response: We will restructure the sentences in the revised manuscript and let a native speaker do the final language editing.

14. Lines 226- 227: It would be more correct to say: " We assume that these deviations were instrument specific and the fitting parameters have to be adjusted for every single device.

Response: Corrected in the revised manuscript.

15. Lines 243-245. I can't see that for the top 4 to 12cm. Clarify please.

Response: Yes, for the calcareous soil there was no increase in $CO_2$ concentration between 4 and 12 cm which is also related to the relative 13C depletion in 4 cm compared to 12 cm – both is assumed to be due to mixing in of atmospheric air (having lower $CO_2$ concentrations and a d13C of approx. -8). We will clarify that in the revised version of the manuscript.

16. Line 246: ...relative to what? Soil $\delta$13CO2 was only slightly enriched, according to Fig. 8

Response: The $\delta$13C signal of soil CO2 at 4 cm depth is enriched compared to the one

sampled from 8cm depth and this is visible in Figure.9. We see a constant depletion in 13C of soil CO2 from 80 to 8 cm soil depth – the 4 cm depth does not fit into that trend as we here see compared to 8 cm a slight enrichment.

17. Lines 242-272: For this whole paragraph it would be worth to read the paper of Cerling, 1984, and Bowen, 2004 (see recommended literature).

Response: The whole paragraph will be modified by including relevant information from Cerling, 1984, and Bowen, 2004.

18. Line 250: No specific pattern...Actually the pattern for $\delta$18O is quite similar to that of the $\delta$13C, except for this sharp decline at around 2:00, (which is less visible for the $\delta$13C time course). The authors should comment that, what could be the cause?

Response: It seems like there was a pressure dip during the specific time window. It occurred due to a short time technical issue rather than any biological process. It can be that the internal pump was not drawing enough gas into the optical cavity there by creating an under pressure in the cavity which then resulted in aberrant values.

19. Line 254: It would be highly beneficial for this statement if you had the $\delta$ values of the soil organic matter for the respective soil depths.

Response: We have measured soil samples for bulk $\delta$13C, bicarbonate $\delta$13C & $\delta$18O values and also % of total carbon in the soil across a depth profile of (0-80 cm) for the calcareous soil (See Fig.4a-c and Fig.5). We observed a slight increase in $\delta$13C values for bulk soil in deeper soil layers (See Fig.4 a,c). Moreover, also the carbonate d13C gets more positive in the 60-80 cm layer. Since total organic carbon content decreases with depth it can be assumed that CO2 derived from carbonate weathering having less negative d13C more strongly contributed to the soil CO2 in this depths (especially since we see an increase in soil CO2 concentration with depth). This is accordance with the laser-based measurements which shows a strong increase in d13C of soil CO2 in the deepest soil layer leading us to the hypothesis that this signal is indicating carbonate

derived CO2.

20. Line 264: It would be more accurate to say: "...is assumed to be the dominating source of soil CO2..."

Response: Corrected in the revised manuscript.

21. Lines 269-272: Are you sure that the $\delta$18O values of the soil CO2 are referred to VSMOW? It looks more like VPDB. Please check that! Then, compared to the $\delta$18O values close to the soil surface CO2 the $\delta$18O values in -80 cm depth are surprisingly high relative to the topsoil. Soil surface water is more prone to be enriched, due to soil surface evaporation processes, than water close to ground water. The authors should comment on that.

Response: $\delta$18O values are reported against VPDB and not VSMOW. We will clarify that in the revised manuscript. When we assume that in 80 cm soil depth a relatively large part of the CO2 derives from carbonate this could explain the strongly enriched 18O signal. We, however, need then to assume that that the oxygen in the CO2 is not in full equilibrium with the precipitation influenced soil water. As mainly microbial carbonic anhydrase mediates the fast equilibrium between CO2 and water in the soil and the microbial activity is low in deeper soil layers (e.g. Schmidt MWI, Torn MS, Abiven S, et al. Persistence of soil organic matter as an ecosystem property. Nature. 2011;478(7367):49-56. doi:10.1038/nature10386.) we can speculate that in deep layers with a significant production of carbonate derived CO2 a lack of full equilibration might be the reason for the observed d18O values.

22. Lines 281-283: Here it would be valuable to have more information on the soil structure. Isn't the acidic soil less compact and dense than the calcareous soil and therefore the diffusivity would be higher in the acidic soil. Its higher CO2 concentration could as well be a result of a higher microbial activity due to its higher organic content. It would be interesting to see soil respiration data for these soils. Maybe the authors can comment on that

Response: Calcareous soil sampled from our study site was gravel rich and less compact. while the acidic soil was more fine, homogeneous and compact. It is sound to consider gas diffusivity in calcareous soil (in our study site) higher in comparison to the acidic soil. It is highly likely that it is due to an increased microbial activity in the acidic soil. We have soil respiration data for the acidic but not for the calcareous soil so we cannot make a comparison.

23. Lines 285-287: Again, are these $\delta$18O values really referring to the VSMOW scale? Then somehow your calculation between the $\delta$18O of the soil water and that of the CO2 is strange. If you add 41‰ (oxygen fractionation between water and CO2) to - 10‰ ($\delta$18O of the soil water) that would result in ca. 31‰ but you indicate -10‰Please clarify.

Response: $\delta$18O values are reported against VPDB and not VSMOW. Will be corrected in the revised manuscript.

24. Conclusion: The first 8 lines are more a summary than a conclusion. Focus on the main outcome of your study, which is the non-linear response of the $\delta$-values versus CO2 concentration. This is a strong demonstration for how essential a careful concentration vs. Isotope ratio calibration is especially when the system is used for such a wide concentration range. Then it would be interesting if your tube-soil-CO2-capture method is reliable and highlight the advantages and disadvantages versus other methods. You practically ignored this method in the discussion. It would be interesting to know more about your experience with it. In that light what do you conclude from your first results?

Response: We agree with the reviewer regarding the fact that the calibration procedure is not well discussed and needs to shed more light into it. We will certainly consider this suggestion and modify the discussion and conclusion parts by including more details and knowledge gained from calibrating the Laser spectrometer in the revised manuscript.

25. Figures: In all Figures, where you plot $\delta18O$ values, check whether you used the VSMOW or VPDB scale.

Response: Yes, we will check and correctly indicate the reference in a revised manuscript.

26. Fig. 1: the expression "water bath" is misleading better to use an expression like "gas thermostat system" or something alike. Clarify whether you used PTFE (brand, type, producer etc.) or Swagelok filters.

Response: "This correction will be made in a revised manuscript. And we used Swagelok filter (Stainless Steel In-Line Particulate Filter, 6 mm Swagelok Tube Fitting, 15 Micron Pore Size)

27. Fig 5 and Fig 6: it would be better to use D-$\delta$ or Diff-$\delta$ instead of $\Delta\delta$

Response: Diff-$\delta$ will be used instead of $\Delta\delta$ in the revised manuscript.

28. Fig. 8: Indicate in the figure legend that this is a "... Time course of the evolution of ..." with the specific time resolution.

Response: Will be included in a revised manuscript.

29. Fig. 9: Indicate in the figure legend that you display "...Daily? averages of CO2 concentration and isotope values in depth profiles..."

Response: The data displayed is an average of measurements taken over 4-hour time period. Will be corrected in the revised manuscript.

Figure.1: Measured $\delta13C$ and $\delta18O$ of CO2 compared to the target values (a,b) when diluted using synthetic air. (c-d) shows the deferences from the target values (diff-$\delta13C$, diff-$\delta18O$) across a concentration gradient. Red and Blue dots show measured $\delta13C$ & $\delta18O$ values of two different gases with distinct isotopic signatures, red and blue dashed lines represent the $\delta13C$ & $\delta18O$ target values of the respective gases calibrated independently by isotope ratio mass spectrometry. Black line denotes model

fit for diff- $\delta$13C, diff-$\delta$18O values across changing CO2 concentration (300 – 25000 ppm).

Figure 2: Residual distribution of modeled data for the differences in d18O between measured and target values (diff- $\delta$13C, diff-$\delta$18O) values across changing CO2 concentration (300 – 25000 ppm).

Figure 3: Corrected $\delta$13C & $\delta$18O values of two different standard gases measured after correcting for concentration dependent drift. The dashed lines indicated the target $\delta$13C and $\delta$180 target values calibrated independently by isotope ratio mass spectrometry.

Figure.4: Bulk $\delta$13C (a), bicarbonate $\delta$13C (c) and % of total carbon (b) in soil across a depth profile of (0-80 cm).

Figure.5: bicarbonate $\delta$18O in soil across a depth profile of (0-80 cm).

Figure.6: Gas temperature recorded inside the optical cavity (Blue line) & Temperature recorded in the thermostat system (black lines).

Figure.7: The figure shows pressure inside the optical cavity (blue line) plotted on right y axis and change in CO2 concentration (black lines) plotted on left y axis. Data was taken while the system is running in a closed loop system with periodic injections of CO2 gas.

Please also note the supplement to this comment:
https://www.soil-discuss.net/soil-2018-9/soil-2018-9-AC2-supplement.pdf

Fig.1

a) [Plot with y-axis δ¹³C (‰) ranging from -60 to 10, showing Gas A (blue) and Gas B (red) data points with Absolute dashed lines]

b) [Plot with y-axis δ¹⁸O (‰) ranging from -70 to 10, showing Gas A (blue) and Gas B (red) data points with Absolute dashed lines]

Legend:
- Gas A (blue dot)
- Gas B (red dot)
- Absolute (red dashed)
- Absolute (blue dashed)
- Model (black solid)

c) [Plot with y-axis Diff-δ¹³C (‰) ranging from -9 to 12, x-axis $CO_2$[ppm] from 0 to 25000]

d) [Plot with y-axis Diff-δ¹⁸O (‰) ranging from 0 to 40, x-axis $CO_2$[ppm] from 0 to 25000]

**Fig. 1.**

[Figure]

[Figure]

**Fig. 2.**

[Figure]

Fig. 3.

[Figure]

[Figure]

**Fig. 4.**

[Figure]

Fig.5

Fig. 5.

[Figure]

Fig.6

Fig. 6.

[Figure]

Fig. 7.

**Supplement:**

1. **Preparation of the calibration gases:** You mixed the gases in N2. This will cause some shifts in your absorption spectra and will result in a shift of your isotopic values as it was shown in Bowling et al. (2003). Tuzon et al., (2008) address the calibration process in detail and it is recommended to consider this paper in this study. If the possibility is still given, it might be worth it produce new reference gases with synthetic CO2 free air (20% oxygen and 80% nitrogen) then repeat the calibration of the instrument, compare the results and reassess the results. I am aware that this is an unusual request and almost too much to ask for but it would be worth it.

Response: As per the suggestions made by both of the referees, we have conducted another round of calibration by diluting the CO2 gas using synthetic air (20% oxygen and 80% nitrogen) instead of N2. Diluting the CO2 standard gas with N2 resulted in a standard deviation of 8.1(‰) for $\delta^{13}C$ values and 4.7 (‰) for $\delta^{18}O$ values respectively. Diluting CO2 standard gases with synthetic air resulted in a standard deviation of 6.44(‰) and 6.818(‰) for $\delta^{13}C$ and $\delta^{18}O$ respectively (see Fig.1a-b). With our new calibration curves (see Fig.1 c-d, &Table.1,2), we are able to bring down standard deviation to 0.08(‰) for $\delta^{13}C$ and 0.04(‰) for $\delta^{18}O$ (see Fig.2a-b (residual distribution), Fig.3a-d (Corrected $\delta^{13}C$ and $\delta^{18}O$ values). We will restructure and include new calibration system in the revised version of the manuscript.

[Figure]

Figure.1: Measured $\delta^{13}C$ and $\delta^{18}O$ of $CO_2$ compared to the target values (a,b) when diluted using synthetic air. (c-d) shows the deferences from the target values (diff- $\delta^{13}C$, diff-$\delta^{18}O$) across a concentration gradient. Red and Blue dots show measured $\delta^{13}C$ & $\delta^{18}O$ values of two different gases with distinct isotopic signatures, red and blue dashed lines represent the $\delta^{13}C$ & $\delta^{18}O$ target values of the respective gases calibrated independently by isotope ratio mass spectrometry. Black line denotes model fit for diff- $\delta^{13}C$, diff-$\delta^{18}O$ values across changing CO2 concentration (300 – 25000 ppm).

[Figure]

[Figure]

Figure 2: Residual distribution of modeled data for the differences in d18O between measured and target values (diff- $\delta^{13}C$, diff-$\delta^{18}O$) values across changing CO2 concentration (300 – 25000 ppm).

[Figure]

Figure 3: Corrected $\delta^{13}C$ & $\delta^{18}O$ values of two different standard gases measured after correcting for concentration dependent drift. The dashed lines indicated the target $\delta^{13}C$ and $\delta^{18}0$ target values calibrated independently by isotope ratio mass spectrometry.

2. How did you calibrate the gases, via gas bench-IRMS or via cryo extraction and Dual Inlet IRMS? If you used the gas bench method how did you handle the problem with the septa of the vacutainers leading to a large scatter for the 18O/16O ratio, in case you used this method?

Response: The $\delta^{13}C$ and $\delta^{18}O$ values of our inhouse calibration gas standards were measured via cryo extraction and Dual Inlet IRMS. This will be included in a revised version of the manuscript.

3. It would be worth to insert subtitles in chapter 3: e.g. 3.1 Instrument calibration and correction (after Line 192) 3.2 Variation in soil CO2 concentration and its C and O isotope values (after line 241)

Response: Subtitles will be added in the modified manuscript.

**Specific comments**

4.  Line 140: PTFE or Swagelok filter? Clarify

Response: Swagelok filter (Stainless Steel In-Line Particulate Filter, 6 mm Swagelok Tube Fitting, 15 Micron Pore Size); this information will be added to the revised version.

5.  Lines 141-142: what kind of a filter is this to prevent moisture from getting into the device? What device do you mean? Normally moisture isn´t captured with a filter but much rather with a water trap. But usually commercially available gas is very dry making a water trap dispensable.

Response: The filter is a particulate matter filter and not a moisture filter. It can hold very little amount of liquid water and not water vapor. This will be rectified in the revised manuscript.

6.  Lines 145-146: If you intend to produce a gas with a temperature range from minus! -20°C to +40°C a water bath is certainly not the right choice. Please clarify. Either you used a different cooling liquid or you never went below 0°C.

[Figure]

Response: The reviewer is right, it needs further clarification. We have used a water bath to increase the temperature to higher values than the room temperature. To reduce the temperature below, we immersed gas tubes in liquid Nitrogen kept in an isotherm flask. This information will be included in the revised manuscript.

7. Line 156: Please indicate the concentration steps for the calibration.

Response: We have used 27 concentration points across the range (300-25000 ppm). For more details see table.3.

Table.3

| CO2 ppm | d13C | Stdev data |
|---|---|---|
| 350.93 | -31.28 | 0.04 |
| 453.32 | -31.42 | 0.07 |
| 543.73 | -31.54 | 0.07 |
| 755.35 | -31.87 | 0.03 |
| 852.19 | -31.94 | 0.03 |
| 951.99 | -32.15 | 0.03 |
| 1257.59 | -32.52 | 0.07 |
| 2377.12 | -33.86 | 0.10 |
| 3670.33 | -35.44 | 0.03 |
| 4651.48 | -36.46 | 0.00 |
| 5230.98 | -37.13 | 0.04 |
| 6718.14 | -38.65 | 0.02 |
| 7441.17 | -39.37 | 0.02 |
| 8396.27 | -40.13 | 0.00 |
| 9491.37 | -41.13 | 0.00 |
| 10390.11 | -41.99 | 0.05 |
| 11402.32 | -42.83 | 0.01 |
| 12488.75 | -43.59 | 0.07 |
| 13531.13 | -44.44 | 0.06 |
| 14532.92 | -45.09 | 0.03 |
| 15534.13 | -45.79 | 0.01 |
| 16547.02 | -46.49 | 0.05 |
| 17255.32 | -46.97 | 0.03 |
| 19893.50 | -48.60 | 0.01 |
| 21237.86 | -49.19 | 0.04 |
| 22462.06 | -49.92 | 0.01 |
| 24313.08 | -50.78 | 0.05 |

8. Line 187: How was the pressure regulated in this closed loop? For a proper operation of the laser instrument, the pressure in the cavity cell must be as constant as possible, since only slightest changes in pressure can mimic a change in concentration of all gas species.

[Figure]

Figure.7: The figure shows pressure inside the optical cavity (blue line) plotted on right y axis and change in CO2 concentration (black lines) plotted on left y axis. Data was taken while the system is running in a closed loop system with periodic injections of CO2 gas.

Response: We did not encounter any pressure differences while maintaining a closed loop system. We have cavity pressure data monitored (see Figure.7). We will include this information in the revised version of the manuscript.

9. Line 204: To prevent misunderstandings it is better to write D-δ or Diff-δ instead of Δδ, since Δ is used for discrimination (fractionation) in the isotope literature.

Response: We agree with the reviewer. It will be changed in the revised manuscript.

10. Line 206: rewrite "... The mathematical model with the most fitting to..." write "...the mathematical model with the best fir for ..."

Response: Corrected in the revised manuscript.

11. Line 211: replace "... most fitting model ..." with "... best fit ..."

Response: Corrected in the revised manuscript.

12. Line 221: replace "… better…" with " … the needed…"

Response: Corrected in the revised manuscript.

13. Lines 223 – 231: A native English-speaking person should reassess these lines.

Response: We will restructure the sentences in the revised manuscript and let a native speaker do the final language editing.

14. Lines 226- 227: It would be more correct to say: " We assume that these deviations were instrument specific and the fitting parameters have to be adjusted for every single device.

Response: Corrected in the revised manuscript.

15. Lines 243-245. I can´t see that for the top 4 to 12cm. Clarify please.

Response: Yes, for the calcareous soil there was no increase in CO2 concentration between 4 and 12 cm which is also related to the relative 13C depletion in 4 cm compared to 12 cm – both is assumed to be due to mixing in of atmospheric air (having lower CO2 concentrations and a d13C of approx. -8). We will clarify that in the revised version of the manuscript.

16. Line 246: …relative to what? Soil δ13CO2 was only slightly enriched, according to Fig. 8

Response: The $\delta^{13}C$ signal of soil $CO_2$ at 4 cm depth is enriched compared to the one sampled from 8cm depth and this is visible in Figure.9. We see a constant depletion in 13C of soil CO2 from 80 to 8 cm soil depth – the 4 cm depth does not fit into that trend as we here see compared to 8 cm a slight enrichment.

17. Lines 242-272: For this whole paragraph it would be worth to read the paper of Cerling, 1984, and Bowen, 2004 (see recommended literature).

Response: The whole paragraph will be modified by including relevant information from Cerling, 1984, and Bowen, 2004.

18. Line 250: No specific pattern...Actually the pattern for δ18O is quite similar to that of the δ13C, except for this sharp decline at around 2:00, (which is less visible for the δ13C time course). The authors should comment that, what could be the cause?

Response: It seems like there was a pressure dip during the specific time window. It occurred due to a short time technical issue rather than any biological process. It can be that the internal pump was not drawing enough gas into the optical cavity there by creating an under pressure in the cavity which then resulted in aberrant values.

19. Line 254: It would be highly beneficial for this statement if you had the δ values of the soil organic matter for the respective soil depths.

[Figure]

Figure.4: Bulk $\delta^{13}C$ (a), bicarbonate $\delta^{13}C$ (c) and % of total carbon (b) in soil across a depth profile of (0-80 cm).

[Figure]

Figure.5: bicarbonate $\delta^{18}O$ in soil across a depth profile of (0-80 cm).

Response: We have measured soil samples for bulk $\delta^{13}C$, bicarbonate $\delta^{13}C$ & $\delta^{18}O$ values and also % of total carbon in the soil across a depth profile of (0-80 cm) for the calcareous soil (See Fig.4a-c and Fig.5). We observed a slight increase in $\delta^{13}C$ values for bulk soil in deeper soil layers (See Fig.4 a,c). Moreover, also the carbonate d13C gets more positive in the 60-80 cm layer. Since total organic carbon content decreases with depth it can be assumed that CO2 derived from carbonate weathering having less negative d13C more strongly contributed to the soil CO2 in this depths (especially since we see an increase in soil CO2 concentration with depth). This is accordance with the laser-based measurements which shows a strong increase in d13C of soil CO2 in the deepest soil layer leading us to the hypothesis that this signal is indicating carbonate derived CO2.

20. Line 264: It would be more accurate to say: "...is assumed to be the dominating source of soil CO2..."

Response: Corrected in the revised manuscript.

21. Lines 269-272: Are you sure that the δ18O values of the soil CO2 are referred to VSMOW? It looks more like VPDB. Please check that! Then, compared to the δ18O

values close to the soil surface $CO_2$ the $\delta18O$ values in -80 cm depth are surprisingly high relative to the topsoil. Soil surface water is more prone to be enriched, due to soil surface evaporation processes, than water close to ground water. The authors should comment on that.

Response: $\delta18O$ values are reported against VPDB and not VSMOW. We will clarify that in the revised manuscript. When we assume that in 80 cm soil depth a relatively large part of the $CO_2$ derives from carbonate this could explain the strongly enriched 18O signal. We, however, need then to assume that that the oxygen in the $CO_2$ is not in full equilibrium with the precipitation influenced soil water. As mainly microbial carbonic anhydrase mediates the fast equilibrium between $CO_2$ and water in the soil and the microbial activity is low in deeper soil layers (e.g. Schmidt MWI, Torn MS, Abiven S, et al. Persistence of soil organic matter as an ecosystem property. *Nature*. 2011;478(7367):49-56. doi:10.1038/nature10386.) we can speculate that in deep layers with a significant production of carbonate derived $CO_2$ a lack of full equilibration might be the reason for the observed d18O values..

22. Lines 281-283: Here it would be valuable to have more information on the soil structure. Isn´t the acidic soil less compact and dense than the calcareous soil and therefore the diffusivity would be higher in the acidic soil. Its higher $CO_2$ concentration could as well be a result of a higher microbial activity due to its higher organic content. It would be interesting to see soil respiration data for these soils. Maybe the authors can comment on that

Response: Calcareous soil sampled from our study site was gravel rich and less compact. while the acidic soil was more fine, homogeneous and compact. It is sound to consider gas diffusivity in calcareous soil (in our study site) higher in comparison to the acidic soil.

It is highly likely that it is due to an increased microbial activity in the acidic soil. We have soil respiration data for the acidic but not for the calcareous soil so we cannot make a comparison.

23. Lines 285-287: Again, are these $\delta18O$ values really referring to the VSMOW scale? Then somehow your calculation between the $\delta18O$ of the soil water and that of the $CO_2$ is strange. If you add 41‰ (oxygen fractionation between water and $CO_2$) to -

10‰ (δ18O of the soil water) that would result in ca. 31‰, but you indicate -10‰. Please clarify.

Response: δ18O values are reported against VPDB and not VSMOW. Will be corrected in the revised manuscript.

24. **Conclusion:** The first 8 lines are more a summary than a conclusion. Focus on the main outcome of your study, which is the non-linear response of the δ-values versus CO2 concentration. This is a strong demonstration for how essential a careful concentration vs. Isotope ratio calibration is especially when the system is used for such a wide concentration range. Then it would be interesting if your tube-soil-CO2-capture method is reliable and highlight the advantages and disadvantages versus other methods. You practically ignored this method in the discussion. It would be interesting to know more about your experience with it. In that light what do you conclude from your first results?

Response: We agree with the reviewer regarding the fact that the calibration procedure is not well discussed and needs to shed more light into it. We will certainly consider this suggestion and modify the discussion and conclusion parts by including more details and knowledge gained from calibrating the Laser spectrometer in the revised manuscript.

25. **Figures:** In all Figures, where you plot δ18O values, check whether you used the VSMOW or VPDB scale.

Response: Yes, we will check and correctly indicate the reference in a revised manuscript.

26. Fig. 1: the expression "water bath" is misleading better to use an expression like "gas thermostat system" or something alike. Clarify whether you used PTFE (brand, type, producer etc.) or Swagelok filters.

Response: "This correction will be made in a revised manuscript. And we used Swagelok filter (Stainless Steel In-Line Particulate Filter, 6 mm Swagelok Tube Fitting, 15 Micron Pore Size)

27. Fig 5 and Fig 6: it would be better to use D-δ or Diff-δ instead of Δδ

Response: Diff-δ will be used instead of Δδ in the revised manuscript.

28. Fig. 8: Indicate in the figure legend that this is a "… Time course of the evolution of …" with the specific time resolution.

Response: Will be included in a revised manuscript.

29. Fig. 9: Indicate in the figure legend that you display "…Daily? averages of CO2 concentration and isotope values in depth profiles…"

Response: The data displayed is an average of measurements taken over 4-hour time period. Will be corrected in the revised manuscript.

Table.1

| Equation | y=a*(b-exp(-c*x)) | | | | |
|---|---|---|---|---|---|
| Standard Error | 0.07468171 | | | | |
| Correlation Coeff.(r) | 0.999941 | | | | |
| Coeff.of Determination (r^2) | 0.99988246 | | | | |
| DOF | 54 | | | | |
| AICC | -294.6349 | | | | |
| Parameters | | | | | |
| Value | Std | Err | Range | (95% | confidence) |
| a | 31.007446 | 0.214984 | 30.576428 | to | 31.438463 |
| b | 0.713759 | 0.002376 | 0.708995 | to | 0.718522 |
| c | 0.000043 | 0 | 0.000042 | to | 0.000043 |
| | | | | | |
| Covariance Matrix | | | | | |
| | a | b | c | | |
| a | 8.286768 | 0.088333 | -0.000018 | | |
| b | 0.088333 | 0.001012 | 0 | | |
| c | -0.000018 | 0 | 0 | | |

Table.2

| Equation | y=a*(b^x)*(x^c) | | | | |
|---|---|---|---|---|---|
| Standard Error | 0.04365503 | | | | |
| Correlation Coeff.(r) | 0.999981 | | | | |
| Coeff.of Determination (r^2) | 0.99996128 | | | | |
| DOF | 51 | | | | |
| AICC | -337.04644 | | | | |
| Parameters | | | | | |
| | Value | StdErr | Err | Range (95% confidence) | |
| a | 0.851623 | 0.003025 | 0.84555 | to | 0.857697 |
| b | 0.999928 | 0 | 0.999928 | to | 0.999928 |
| c | 0.477819 | 0.000472 | 0.476871 | to | 0.478767 |
| | | | | | |
| Covariance Matrix | | | | | |
| | a | b | c | | |
| a | 0.004803 | 0 | - 0.000745 | | |
| b | 0 | 0 | 0 | | |
| c | -0.000745 | 0 | 0.000117 | | |

---

## Author Comment (AC3) · 30 Sep 2018

General comment # Remarks to the Author While the introduction section discussing soil CO2 measurement techniques does include some discussion of chamber and tower sampling, it should also include references that made use of soil gas wells to sample CO2 for d13C and d18O values, as that is more similar to the work the paper describes. I suggest you include these references, both of which are relevant to arid and semi-arid soils, as that is the focus of the manuscript under discussion:

Breecker, D., & Sharp, Z. D. (2008). A field and laboratory method for monitoring the concentration and isotopic composition of soil CO2. Rapid Communications in Mass

[Figure]

Spectrometry, 22(4), 449-454.

Oerter, E. J., & Amundson, R. (2016). Climate controls on spatial and temporal variations in the formation of pedogenic carbonate in the western Great Basin of North America. GSA Bulletin, 128(7-8), 1095-1104.

Response: I will include the suggested references as they are apt for the topic of discussion.

---

## Author Response (AR1)

This manuscript describes attempts to calibrate a laser-based absorption instrument for use in high-frequency measurements of 13C and 18O of CO2 in soil depth profiles, and provides very brief field data from two sites. This type of work is useful in that many investigators use new instruments, such as the LGR instrument described here, without sufficient validation. However, I have some major concerns about the calibration method which the authors can hopefully address in a revised manuscript. In addition, the paper would be much stronger if additional field data were presented, especially along with atmospheric measurements at the soil surface, which are needed to calculate the isotope composition of soil respired CO2 (as opposed to soil profile CO2). At present, it appears that <24 hours of field data are shown.

1. The authors develop non-linear calibration functions to account for the concentration dependence of isotope ratios but it is not clear to me how these functions might also vary in isotope space (i.e., as a dual function of isotope composition and concentration). For example, fig. 5 shows the correction functions for concentration dependence but does not show how/if these varied as a function of the isotope ratios of the standard gasses, which should all be shown on this figure. Furthermore, Fig 6 and 7 show serious deviations of calibrated vs. true values for both 13C and 18O between circa 2000 – 10000 ppm, of as much as 2 permil, even though those differences disappear at higher values. This deviation is unacceptable given the requirements of analyzing soil CO2, where differences of 2 permil may be highly significant from an ecological perspective. I note that CO2 concentrations < 10,000 ppm are commonplace in most soil profiles, especially in shallow horizons that typically dominate CO2 production, such that capacity for accurate and precise measurements in this lower concentration range is really critical. Even greater variability is shown in Fig 7, which appears to reach 4 per mil. This is not acceptable for natural abundance work.

Response:  As per the suggestions made by both of the referees, we have conducted another round of calibration by diluting CO2 gas using synthetic air instead of N2. Diluting the CO2 standard gas with N2 resulted in a standard deviation of 8.1(‰) for $\delta^{13}C$ values and 4.7 (‰) for $\delta^{18}O$ values respectively. Diluting CO2 standard gases with synthetic air resulted in a standard deviation of 6.44(‰) and 6.802(‰) for $\delta^{13}C$ and $\delta^{18}O$ respectively (see Fig.1a-b). With our new calibration curves (see Fig.1 c-d, &Table.1,2), we are able to bring down standard deviation to 0.08(‰) for $\delta^{13}C$ and 0.09(‰) for $\delta^{18}O$ (see Fig.2a-b (residual distribution), Fig.3a-d (Corrected $\delta^{13}C$ and $\delta^{18}O$ values). By introducing the new calibration correction (see.Fig.3) the values are very close to the target value across the whole concentration range and hence we are confident that the system is suitable for ecosystem studies based on measuring subtle changes in isotopic signature of CO2 across plant soil atmosphere continuum.

[Figure]

Figure.1: Deviation of measured $\delta^{13}C$ and $\delta^{18}O$ of CO2 (a,b) when diluted using synthetic air. (c-d) shows diff- $\delta^{13}C$, diff-$\delta^{18}O$ values across a concentration gradient. Red and Blue dots shows measured $\delta^{13}C$ & $\delta^{18}O$ values of two different gases with distinct isotopic signatures, red and blue dashed lines represents absolute $\delta^{13}C$ & $\delta^{18}O$ values of the respective gases. Black line denotes model fit for diff- $\delta^{13}C$, diff-$\delta^{18}O$ values across changing CO2 concentration (300 – 25000 ppm).

[Figure]

[Figure]

Figure 2: Residual distribution of modeled data for diff- $\delta^{13}$C, diff-$\delta^{18}$O values across changing CO2 concentration (300 – 25000 ppm).

[Figure]

Figure 3: Corrected $\delta^{13}C$ & $\delta^{18}O$ values of two different standard gases measured after correcting for concentration dependent drift.

2. Also, basic details about the soils investigated are missing that are necessary to interpret the measured values of 13C and 18O of CO2. For example, what are the carbonate concentrations and isotope ratios in the calcareous soil, and how do they vary with depth? What are the 13C values of SOM? This is a prerequisite for interpreting the soil profile CO2 values. Also, to calculate the isotope ratios of soil-respired CO2, we need measurements of the atmospheric boundary condition. See Davidson 1995 GCA, doi:10.1016/0016-7037(95)00143-3. Note that several recent papers neglected have reported 13C of CO2 from soil profiles using high temporal-resolution optical measurements, these should be discussed or at least mentioned.

[Figure]

Figure.4: Bulk $\delta^{13}C$ (a), bicarbonate $\delta^{13}C$ (c) and % of carbon (b) in soil across a depth profile of (0-80 cm).

[Figure]

Figure.5: bicarbonate $\delta^{18}O$ in soil across a depth profile of (0-80 cm).

Response: We have measured soil samples for bulk $\delta^{13}C$, bicarbonate $\delta^{13}C$ & $\delta^{18}O$ values and also % carbon content in the soil across a depth profile of (0-80 cm) for the calcareous soil (See Fig.4a-c and Fig.5). We observed an increase in $\delta^{13}C$ values (of bulk soil and carbonate) in deeper soil layers (See Fig.4 a,c). This fits to our assumption of an increased contribution of bicarbonate derived $\delta^{13}C$ enriched $CO_2$ in deeper soil layers.

Our aim is to establish a method which enables continuous online measurement of soil gas $\delta^{13}C$ & $\delta^{18}O$ values at different soil depths and hence calculating the isotope ratios of soil-respired $CO_2$ is not done in this manuscript. This would be beyond the scope of a calibration focused paper – we however show the importance to properly calibrate laser-based systems to obtain valid measurements of d13C and d18O of soil $CO_2$ which is a prerequisite for assessing the rate and isotopic composition of soil respiration.

3. I am skeptical as to the validity of the temperature tests employed. Note that we need to know the temperature of the analyte gas itself, which may be substantially different than the temperature of the water bath through which it circulates unless the residence time of the gas in the tubing and the heat transfer properties of the tubing allow for sufficiently rapid temperature equilibration, which may not completely occur if flow rates are high. For example, certain applications require heating of gasses at a sampling inlet to avoid condensation, yet the temperature of the gas at the point of the analysis may be substantially different (e.g. -4 – 40C) for some other optical gas analyzers, and should optimally be controlled within the analyzer cavity itself. Thus, unless the exact temperature of the gas at the point of measurement can be determined, I would not trust the results from the water bath experiment. Regardless, details of the analysis flow rate should be reported (and whether these rates were controlled during sample analysesâ˘TMFC's are mentioned for standards only).

Response: The laser spectrometer was able to maintain the temperature inside the optical cavity quite stable at 46.61°C irrespective of the fluctuations in the gas temperature (See Figure attached below). It is clear that the temperature maintained in the water bath will not get directly reflected in the sample gas due to multiple reasons including diffusion barrier of the PTFE tubing and higher flow rates, nevertheless, there will be an increment or decrement in the gas temperature. The aim is to show that the system is also stable in field conditions where temperature fluctuation is happening. The system is running in a closed loop meaning there is enough time for the gas for heat exchange. We have adjusted the part where test for equipment stability under fluctuating temperature is done in the modified manuscript.

[Figure]

Fig.6

4. There are numerous issues with grammar, style, and errant capitalization throughout. The figures and tables have a strange mix of fonts (be consistent!) and the legends are compressed. Please follow standard procedures for presenting your MS (provide captions as text in the document, not as images). There is a significant typo in Table 1.

Response: This is addressed and rectified in the revised manuscript. We have also let a native speaker do the final editing of the manuscript.

5. Finally, it should be noted that the useful temporal resolution of the measurements will never actually be 1hz as reported given the Allan variance results.

Response: Not exactly clear what the reviewer meant by "it should be noted that the useful temporal resolution of the measurements will never actually be 1hz as reported given the Allan variance results". It is always useful to get a better temporal resolution which can be used for identifying short term dynamics of CO2 efflux (e.g., diurnal pattern of soil CO2 efflux). Meaning more data points are available for taking an average across a time frame which is best for noise correction by using Allan deviation curves.

6. Was water vapor removed from the analyte gas, and if so, how?

Response: Yes, water vapor was removed using drierite desiccant cartridges. We have now add this information to the revised manuscript.

Table.1

| Equation | y=a*(b-exp(-c*x)) | | | | | |
|---|---|---|---|---|---|---|
| Standard Error | 0.07468171 | | | | | |
| Correlation Coeff.(r) | 0.999941 | | | | | |
| Coeff.of Determination (r^2) | 0.99988246 | | | | | |
| DOF | 54 | | | | | |
| AICC | -294.6349 | | | | | |
| Parameters | | | | | | |
| Value | Std | Err | Range | (95% | | confidence) |
| a | 31.007446 | 0.214984 | 30.576428 | to | | 31.438463 |
| b | 0.713759 | 0.002376 | 0.708995 | to | | 0.718522 |
| c | 0.000043 | 0 | 0.000042 | to | | 0.000043 |
| | | | | | | |
| Covariance Matrix | | | | | | |
| | a | b | c | | | |
| a | 8.286768 | 0.088333 | -0.000018 | | | |
| b | 0.088333 | 0.001012 | 0 | | | |
| c | -0.000018 | 0 | 0 | | | |

Table.2

| Equation | y=a*(b^x)*(x^c) | | | | |
|---|---|---|---|---|---|
| Standard Error | 0.04365503 | | | | |
| Correlation Coeff.(r) | 0.999981 | | | | |
| Coeff.of Determination (r^2) | 0.99996128 | | | | |
| DOF | 51 | | | | |
| AICC | -337.04644 | | | | |
| Parameters | | | | | |
| | Value | StdErr | Err | Range (95% confidence) | |
| a | 0.851623 | 0.003025 | 0.84555 | to | 0.857697 |
| b | 0.999928 | 0 | 0.999928 | to | 0.999928 |
| c | 0.477819 | 0.000472 | 0.476871 | to | 0.478767 |
| | | | | | |
| Covariance Matrix | | | | | |
| | a | b | c | | |
| a | 0.004803 | 0 | - 0.000745 | | |
| b | 0 | 0 | 0 | | |
| c | -0.000745 | 0 | 0.000117 | | |

1. **Preparation of the calibration gases:** You mixed the gases in N2. This will cause some shifts in your absorption spectra and will result in a shift of your isotopic values as it was shown in Bowling et al. (2003). Tuzon et al., (2008) address the calibration process in detail and it is recommended to consider this paper in this study. If the possibility is still given, it might be worth it produce new reference gases with synthetic CO2 free air (20% oxygen and 80% nitrogen) then repeat the calibration of the instrument, compare the results and reassess the results. I am aware that this is an unusual request and almost too much to ask for but it would be worth it.

   Response: As per the suggestions made by both of the referees, we have conducted another round of calibration by diluting the CO2 gas using synthetic air (20% oxygen and 80% nitrogen) instead of N2. Diluting the CO2 standard gas with N2 resulted in a standard deviation of 8.1(‰) for $\delta^{13}C$ values and 4.7 (‰) for $\delta^{18}O$ values respectively. Diluting CO2 standard gases with synthetic air resulted in a standard deviation of 6.44(‰) and 6.802(‰) for $\delta^{13}C$ and $\delta^{18}O$ respectively (see Fig.1a-b). With our new calibration curves (see Fig.1 c-d, &Table.1,2), we are able to bring down standard deviation to 0.08(‰) for $\delta^{13}C$ and 0.09(‰) for $\delta^{18}O$ (see Fig.2a-b (residual distribution), Fig.3a-d (Corrected $\delta^{13}C$ and $\delta^{18}O$ values). We now have restructured the respective section in the manuscript and have included the new calibration system in the revised version of the manuscript.

   "**S**tandard gases were released to a mass flow controller (ANALYT-MTC, series 358, MFC1) after passing through a pressure controller valve (See Figure. 1) with safety bypass (TESCOM, D43376-AR-00-X1-S; V5). A Swagelok filter, ((Stainless Steel All-Welded In-Line Filter (Swagelok, SS-4FWS-05; F1)) was installed at the inlet of the flow controller (ANALYT-MTC, series 358; MFC1). **Synth-air** was released and passed to another flow controller (ANALYT-MTC, series 358; MFC2) through a Swagelok filter (F2 in Figure. 1). CO2 and **synth-air** leaving the flow controllers (MFC1 and MFC2 respectively) were then mixed and drawn through a ¼" Teflon tube (P8), which was kept in a **gas thermostat unit** (See Figure.1). *The **thermostat unit** contained, a) a thermostat-controlled water bath (Kottermann, 3082) and b) an Isotherm flask containing liquid nitrogen. The water bath was used to raise the temperature above room temperature and also to bring the temperature down to +5°C, by placing ice packs in the water bath. To reach low temperatures (-20°C), we immersed the tubes in the isotherm flask filled with liquid N2.* Leaving the **thermostat unit**, the gas was directed to the multiport inlet unit of the OA-ICOS. By using the thermostat unit, we introduced a shift in the reference gas temperature and the aim was to test the temperature sensitivity of the OA-ICOS in measuring $\delta^{13}C$ and $\delta^{18}O$ values. The third CO2 standard gas (which is used for validation) was produced by mixing the other two gas standards in equal molar proportions in a 10L volume plastic bag with inner aluminum foil coating and welded seams (CO2 mix: Linde PLASTIGAS®) under 0.03 MPa pressure by diluting to the required concentration using **synth-air**. The mixture was then temperature adjusted and delivered to the multiport inlet unit (MIU) by using a ¼" Teflon tube (P10). From the multiport inlet unit, calibration gases were delivered into the OA-ICOS for measurement using a ¼" Teflon tube (P9) at a pressure < 0.0689 MPa, with a flow rate of 500 mL/min. The gas leaving the OA-ICOS through the exhaust was fed back to the ¼" Teflon tube (P8) by using a Swagelok pipe Tee (Stainless Steel Pipe Fitting, Male Tee, ¼". Male NPT), intersecting P8 line before entering the **thermostat unit**. Thus, the gas fed was looped in the system until steady values were reported by the OA-ICOS based on $CO_2$ [ppm], $\delta^{13}C$ and $\delta^{18}O$ measurements. *$CO_2$ gas standards were measured at 27 different $CO_2$ concentration levels ranging between 400 and 25,000 ppm.* Every hour before sampling, **synth-air** gas was flushed through the system to remove $CO_2$ to avoid memory effects. The calibration gases were measured in a sequence one after the other four times. During each round of measurement, every calibration gas was diluted to different concentrations of $CO_2$ (400 - 25,000 ppm) and the respective isotopic signature and concentration were determined."

[Figure]

Figure.1: Measured $\delta^{13}C$ and $\delta^{18}O$ of $CO_2$ compared to the target values (a,b) when diluted using synthetic air. (c-d) shows the differences from the target values (diff- $\delta^{13}C$, diff-$\delta^{18}O$) across a concentration gradient. Red and Blue dots show measured $\delta^{13}C$ & $\delta^{18}O$ values of two different gases with distinct isotopic signatures, red and blue dashed lines represent the $\delta^{13}C$ & $\delta^{18}O$ target values of the respective gases calibrated independently by isotope ratio mass spectrometry. Black line denotes model fit for diff- $\delta^{13}C$, diff-$\delta^{18}O$ values across changing CO2 concentration (300 – 25000 ppm).

[Figure]

[Figure]

Figure 2: Residual distribution of modeled data for the differences in d18O between measured and target values (diff- $\delta^{13}$C, diff-$\delta^{18}$O) values across changing CO2 concentration (300 – 25000 ppm).

[Figure]

Figure 3: Corrected $\delta^{13}C$ & $\delta^{18}O$ values of two different standard gases measured after correcting for concentration dependent drift. The dashed lines indicated the target $\delta^{13}C$ and $\delta^{18}O$ target values calibrated independently by isotope ratio mass spectrometry.

2. How did you calibrate the gases, via gas bench-IRMS or via cryo extraction and Dual Inlet IRMS? If you used the gas bench method how did you handle the problem with the septa of the vacutainers leading to a large scatter for the 18O/16O ratio, in case you used this method?

   Response: The $\delta^{13}C$ and $\delta^{18}O$ values of our inhouse calibration gas standards were measured via cryo extraction and Dual Inlet IRMS. This information is now included in the revised version of the manuscript.

   "The $\delta^{13}C$ and $\delta^{18}O$ values of our inhouse calibration gas standards were measured via cryo-extraction and Dual Inlet IRMS"

3. It would be worth to insert subtitles in chapter 3: e.g. 3.1 Instrument calibration and correction (after Line 192) 3.2 Variation in soil CO2 concentration and its C and O isotope values (after line 241)

Response: Subtitles are added in the modified manuscript.

**Specific comments**

4. Line 140: PTFE or Swagelok filter? Clarify

   Response: Swagelok filter (Stainless Steel In-Line Particulate Filter, 6 mm Swagelok Tube Fitting, 15 Micron Pore Size); this information is added to the revised version.

   "A Swagelok filter, ((Stainless Steel All-Welded In-Line Filter (Swagelok, SS-4FWS-05; F1)) was installed at the inlet of the flow controller (ANALYT-MTC, series 358; MFC1). Synth-air was released and passed to another flow controller (ANALYT-MTC, series 358; MFC2) through a Swagelok filter (F2 in Figure. 1)."

5. Lines 141-142: what kind of a filter is this to prevent moisture from getting into the device? What device do you mean? Normally moisture isn´t captured with a filter but much rather with a water trap. But usually commercially available gas is very dry making a water trap dispensable.

   Response: The filter is a particulate matter filter and not a moisture filter. It can hold very little amount of liquid water and not water vapor. This is rectified in the revised manuscript.

6. Lines 145-146: If you intend to produce a gas with a temperature range from minus! -20°C to +40°C a water bath is certainly not the right choice. Please clarify. Either you used a different cooling liquid or you never went below 0°C.

[Figure]

Fig.6: Gas temperature recorded inside the optical cavity (Blue line) & Temperature recorded in the thermostat system (black lines).

Response: The reviewer is right, it needs further clarification. We have used a water bath to increase the temperature to higher values than the room temperature. To reduce the temperature below, we immersed gas tubes in liquid Nitrogen kept in an isotherm flask. This information is included in the revised manuscript.

"*The **thermostat unit** contained, a) a thermostat-controlled water bath (Kottermann, 3082) and b) an Isotherm flask containing liquid nitrogen. The water bath was used to raise the temperature above room temperature and also to bring the temperature down to +5°C, by placing ice packs in the water bath. To reach low temperatures (-20°C), we immersed the tubes in the isotherm flask filled with liquid $N_2$.* Leaving the **thermostat unit**, the gas was directed to the multiport inlet unit of the OA-ICOS. By using the thermostat unit, we introduced a shift in the reference gas temperature and the aim was to test the temperature sensitivity of the OA-ICOS in measuring $\delta^{13}C$ and $\delta^{18}O$ values."

7. Line 156: Please indicate the concentration steps for the calibration.

Response: We have used 27 concentration points across the range (300-25000 ppm). For more details see table.3.

Table.3

| CO2 ppm | d13C | Stdev data |
|---|---|---|
| 350.93 | -31.28 | 0.04 |
| 453.32 | -31.42 | 0.07 |
| 543.73 | -31.54 | 0.07 |
| 755.35 | -31.87 | 0.03 |
| 852.19 | -31.94 | 0.03 |
| 951.99 | -32.15 | 0.03 |
| 1257.59 | -32.52 | 0.07 |
| 2377.12 | -33.86 | 0.10 |
| 3670.33 | -35.44 | 0.03 |
| 4651.48 | -36.46 | 0.00 |
| 5230.98 | -37.13 | 0.04 |
| 6718.14 | -38.65 | 0.02 |
| 7441.17 | -39.37 | 0.02 |
| 8396.27 | -40.13 | 0.00 |
| 9491.37 | -41.13 | 0.00 |
| 10390.11 | -41.99 | 0.05 |
| 11402.32 | -42.83 | 0.01 |

| | | |
|---|---|---|
| 12488.75 | -43.59 | 0.07 |
| 13531.13 | -44.44 | 0.06 |
| 14532.92 | -45.09 | 0.03 |
| 15534.13 | -45.79 | 0.01 |
| 16547.02 | -46.49 | 0.05 |
| 17255.32 | -46.97 | 0.03 |
| 19893.50 | -48.60 | 0.01 |
| 21237.86 | -49.19 | 0.04 |
| 22462.06 | -49.92 | 0.01 |
| 24313.08 | -50.78 | 0.05 |

8.  Line 187: How was the pressure regulated in this closed loop? For a proper operation of the laser instrument, the pressure in the cavity cell must be as constant as possible, since only slightest changes in pressure can mimic a change in concentration of all gas species.

[Figure]

Figure.7: The figure shows pressure inside the optical cavity (blue line) plotted on right y axis and change in CO2 concentration (black lines) plotted on left y axis. Data was taken while the system is running in a closed loop system with periodic injections of CO2 gas.

Response: We did not encounter any pressure differences while maintaining a closed loop system. We have cavity pressure data monitored (see Figure.7). We have included this information in the revised version of the manuscript.

"We experienced no drop in cavity pressure while maintaining a closed loop (See Supplementary Figure S2)."

9. Line 204: To prevent misunderstandings it is better to write D-δ or Diff-δ instead of Δδ, since Δ is used for discrimination (fractionation) in the isotope literature.

Response: We agree with the reviewer. It is corrected in the revised manuscript.

"The dependency of $\delta^{13}C$ and $\delta^{18}O$ values on the $CO_2$ concentration was compensated by using a nonlinear model. The deviations (Diff-δ) of the measured delta values ($\delta_{(OA\text{-}ICOS)}$) from the absolute value of the standard gas ($\delta_{(IRMS)}$) at different concentrations of $CO_2$ were calculated (Diff-δ = $\delta_{(OA\text{-}ICOS)}$ - $\delta_{(IRMS)}$). Several mathematical models were then fitted on Diff-δ as a function of changing $CO_2$ concentration (See figure.5). The mathematical model with the best fit for Diff-δ data was selected using Akaike information criterion corrected (AICc) (Glatting et al., 2007; Hurvich and Tsai, 1989; Yamaoka et al., 1978). The non-linear model fits applied for Diff-$\delta^{13}C$, and Diff-$\delta^{18}O$ measurements are given in Tables 1 & 2, respectively. For Diff-$\delta^{13}C$, a three-parameter exponential model fitted best with $r^2 = 0.99$ (see Table 3 for the values of the parameters, see supplementary Figure S3 (a) for model residuals), and a three-parameter power function model (see Table 2) with $r^2 = 0.99$ showed the best fit for Diff-$\delta^{18}O$ (see Table 3 for the values of the parameters, see supplementary Figure S3 (b) for model residuals)."

10. Line 206: rewrite "… The mathematical model with the most fitting to…" write "…the mathematical model with the best fir for …"

Response: Corrected in the revised manuscript.

"The mathematical model with the best fit for Diff-δ data was selected using Akaike information criterion corrected (AICc)"

11. Line 211: replace "… most fitting model …" with "… best fit …"

Response: Corrected in the revised manuscript.

"The best fit was then introduced into the measured isotopic data ($\delta^{13}C$ and $\delta^{18}O$) and corrected for concentration-dependent errors (See figure. 6)"

12. Line 221: replace "… better…" with " … the needed…"

Response: Corrected in the revised manuscript.

"we found that routine calibration (Correction for concentration-dependent error plus three-point calibration) was inevitable for obtaining the required accuracy, in particular under fluctuating $CO_2$ concentrations."

13. Lines 223 – 231: A native English-speaking person should reassess these lines.

Response: A native speaker has now seen the manuscript for final language editing.

14. Lines 226- 227: It would be more correct to say: " We assume that these deviations were instrument specific and the fitting parameters have to be adjusted for every single device.

Response: Corrected in the revised manuscript.

"We assume that these deviations are instrument specific and the fitting parameters need to be adjusted for every single device"

15. Lines 243-245. I can´t see that for the top 4 to 12cm. Clarify please.

Response: Yes, for the calcareous soil there was no increase in CO2 concentration between 4 and 12 cm which is also related to the relative 13C depletion in 4 cm compared to 12 cm – both is assumed to be due to mixing in of atmospheric air (having lower CO2 concentrations and a d13C of approx. -8). We have clarified that in the revised version of the manuscript.

"Relative $^{13}$C enrichment of the $CO_2$ in the topsoil (4 cm) compared to 8 cm depth is probably due to the invasive diffusion of atmospheric $CO_2$ which has a $\delta^{13}$C value close to -8‰ (e.g., (Levin et al., 1995) )."

16. Line 246: …relative to what? Soil δ13CO2 was only slightly enriched, according to Fig. 8

Response: The $\delta^{13}$C signal of soil $CO_2$ at 4 cm depth is enriched compared to the one sampled from 8cm depth and this is visible in Figure.9. We see a constant depletion in 13C of soil CO2 from 80 to 8 cm soil depth – the 4 cm depth does not fit into that trend as we here see compared to 8 cm a slight enrichment.

17. Lines 242-272: For this whole paragraph it would be worth to read the paper of Cerling, 1984, and Bowen, 2004 (see recommended literature).

Response: The whole paragraph is modified by including relevant information from Cerling, 1984, and Bowen, 2004.

"The patterns observed for the $\delta^{13}$C values of $CO_2$ in the calcareous soil with $^{13}$C enrichment in deeper soil layers can be explained by a substantial contribution of $CO_2$ from abiotic origin to total soil $CO_2$ release as a result of carbonate weathering and subsequent out-gassing from soil water (Schindlbacher et al., 2015).

According to Cerling (1984), the distinct oxygen and carbon isotopic composition of soil carbonate depends on the isotopic signature of meteoric water and to the proportion of $C_4$ biomass present at the time of carbonate formation (Cerling, 1984). $CO_2$ released as a result from carbonates have a distinct $\delta^{13}C$ value close to 0‰ vs. VPDB, while $CO_2$ released during biological respiratory processes has usually $\delta^{13}C$ values around -24‰ as observed in the acidic soil (Figure 10 (e)). Even though the contribution of $CO_2$ from abiotic sources to soil $CO_2$ is often considered to be low, several studies have reported significant proportions ranging between (10 - 60%) emanating from abiotic sources (Emmerich, 2003; Plestenjak et al., 2012; Ramnarine et al., 2012; Serrano-Ortiz et al., 2010; Stevenson and Verburg, 2006; Tamir et al., 2011). Bowen and Beerling, (2004) showed that isotope effects associated with soil organic matter decomposition can cause a strong gradient in $\delta$ values of soil organic matter (SOM) with depth, but are not always reflected in the $\delta^{13}C$ values of soil $CO_2$. We have measured soil samples for bulk soil $\delta^{13}C$, bicarbonate $\delta^{13}C$ & $\delta^{18}O$ values and also determined the percentage of total carbon in the soil across a depth profile of (0-80 cm) (See Figure 8). We observed an increase in $\delta^{13}C$ values for bulk soil in deeper soil layers (See Figure 8 (a,c))."

18. Line 250: No specific pattern...Actually the pattern for δ18O is quite similar to that of the δ13C, except for this sharp decline at around 2:00, (which is less visible for the δ13C time course). The authors should comment that, what could be the cause?

Response: We have now added the following section to explain this pattern: " For $\delta^{18}O$ values of calcareous soil, the depth profile showed no specific pattern except for the $\delta^{18}O$ values at 80 cm depth was found to be less negative. The $\delta^{18}O$ value in the top 4 cm was found to be slightly more enriched that the 8 cm depth and between 8 cm – 35 cm, $\delta^{18}O$ values showed little variation relative to each other. For the sub-daily measurements, we observed a sharp decline in $\delta^{18}O$ values at around 02:00, which is also observed but less pronounced for $\delta^{13}C$ signal. We assume that, the reason for such aberrant values is rather a technical issue than any biological process. It can be due to the fact that the internal pump in the OA-ICOS was not taking adequate amount of gas into the optical cavity, thereby creating a negative pressure inside the cavity resulting in the observed aberrant values."

19. Line 254: It would be highly beneficial for this statement if you had the δ values of the soil organic matter for the respective soil depths.

[Figure]

Figure.4: Bulk δ¹³C (a), bicarbonate δ¹³C (c) and % of total carbon (b) in soil across a depth profile of (0-80 cm).

[Figure]

Figure.5: bicarbonate δ¹⁸O in soil across a depth profile of (0-80 cm).

Response: We have measured soil samples for bulk δ¹³C, bicarbonate δ¹³C & δ¹⁸O values and also % total carbon in the soil across a depth profile of (0-80 cm) for the calcareous soil (See Fig.4a-c and Fig.5). We observed a slight increase in δ¹³C values for bulk soil in deeper soil layers (See Fig.4 a,c). Moreover, also the carbonate d13C gets more positive in the 60-80 cm layer. Since total organic carbon content decreases with depth it can be assumed that CO2 derived from carbonate weathering having less negative d13C more strongly contributed to the soil CO2 in this depth (especially since we see an increase in soil CO2 concentration with depth). This is accordance with the laser-based measurements which shows a strong increase in d13C of soil CO2 in the deepest soil layer leading us to the hypothesis that this signal is indicating carbonate derived CO2.

20. Line 264: It would be more accurate to say: "...is assumed to be the dominating source of soil CO2..."

Response: Corrected in the revised manuscript.

"In contrast to the deeper soil layers, where the carbonate content is high, $CO_2$ from carbonate weathering is assumed to be a dominating source of soil $CO_2$"

21. Lines 269-272: Are you sure that the $\delta 18O$ values of the soil CO2 are referred to VSMOW? It looks more like VPDB. Please check that! Then, compared to the $\delta 18O$ values close to the soil surface CO2 the $\delta 18O$ values in -80 cm depth are surprisingly high relative to the topsoil. Soil surface water is more prone to be enriched, due to soil surface evaporation processes, than water close to ground water. The authors should comment on that.

Response: $\delta 18O$ values are reported against VPDB and not VSMOW. This is corrected in the revised manuscript. When we assume that in 80 cm soil depth a relatively large part of the CO2 derives from carbonate this could explain the strongly enriched 18O signal. We, however, need then to assume that that the oxygen in the CO2 is not in full equilibrium with the precipitation influenced soil water. As mainly microbial carbonic anhydrase mediates the fast equilibrium between CO2 and water in the soil and the microbial activity is low in deeper soil layers (e.g. Schmidt MWI, Torn MS, Abiven S, et al. Persistence of soil organic matter as an ecosystem property. *Nature*. 2011;478(7367):49-56. doi:10.1038/nature10386.) we can speculate that in deep layers with a significant production of carbonate derived CO2 a lack of full equilibration might be the reason for the observed d18O values.

22. Lines 281-283: Here it would be valuable to have more information on the soil structure. Isn't the acidic soil less compact and dense than the calcareous soil and therefore the diffusivity would be higher in the acidic soil. Its higher CO2 concentration could as well be a result of a higher microbial activity due to its higher organic content. It would be interesting to see soil respiration data for these soils. Maybe the authors can comment on that

Response: Calcareous soil sampled from our study site was gravel rich and less compact. while the acidic soil was more fine, homogeneous and compact. It is sound to consider gas diffusivity in calcareous soil (in our study site) higher in comparison to the acidic soil.

It is highly likely that it is due to an increased microbial activity in the acidic soil. We have soil respiration data for the acidic but not for the calcareous soil so we cannot make a comparison.

23. Lines 285-287: Again, are these δ18O values really referring to the VSMOW scale? Then somehow your calculation between the δ18O of the soil water and that of the CO2 is strange. If you add 41‰ (oxygen fractionation between water and CO2) to - 10‰ (δ18O of the soil water) that would result in ca. 31‰, but you indicate -10‰. Please clarify.

Response: δ18O values are reported against VPDB and not VSMOW. This is corrected in the revised manuscript as follows:

"Assuming an $^{18}O$ fractionation of 41‰ between $CO_2$ and water (Brenninkmeijer et al., 1983) this would result in an expected value for $CO_2$ of $\approx$ -10 ± 2‰ vs. VPDB-$CO_2$"

24. **Conclusion:** The first 8 lines are more a summary than a conclusion. Focus on the main outcome of your study, which is the non-linear response of the δ-values versus CO2 concentration. This is a strong demonstration for how essential a careful concentration vs. Isotope ratio calibration is especially when the system is used for such a wide concentration range. Then it would be interesting if your tube-soil-CO2-capture method is reliable and highlight the advantages and disadvantages versus other methods. You practically ignored this method in the discussion. It would be interesting to know more about your experience with it. In that light what do you conclude from your first results?

Response: We agree with the reviewer regarding the fact that the calibration procedure is not well discussed and needs to shed more light into it. We have now rewritten the conclusion section to focus on the main outcome of the study.

"During our preliminary tests with the OA-ICOS, we found that the equipment was highly sensitive to changes in $CO_2$ concentrations. We found a non-linear response of the $\delta^{13}C$ and $\delta^{18}O$ values against changes in $CO_2$ concentration. Given the fact that laser-based $CO_2$ isotope analyzers are getting deployed more commonly in tracing various ecosystem processes, we think that it is important to address this issue.

Therefore, we developed a calibration strategy for correcting errors introduced in $\delta^{13}C$ and $\delta^{18}O$ measurements due to the sensitivity of the device against changing $CO_2$ concentrations. We found that the OA-ICOS measures stable isotopes of $CO_2$ gas samples with a precision comparable to conventional IRMS. The method described in this work for measuring $CO_2$ concentration, $\delta^{13}C$ and $\delta^{18}O$ values in soil air profiles using an OA-ICOS and hydrophobic gas permeable tubes are promising and can be applied for soil $CO_2$ flux studies. As this set up is capable of measuring continuously for longer time periods at higher temporal resolution (1 Hz), it offers greater potential to investigate the isotopic identity of $CO_2$ and the interrelation between soil $CO_2$ and soil water. By using our measurement setup, we could identify abiotic as well as biotic contributions to the soil $CO_2$ in the calcareous soil. We infer that that degassing of $CO_2$ from carbonates due to weathering and evasion of $CO_2$ from groundwater may leave the soil $CO_2$ with a specific and distinct $\delta^{13}C$ signature especially when the biotic activity is rather low."

25. **Figures:** In all Figures, where you plot δ18O values, check whether you used the VSMOW or VPDB scale.

Response: Yes, all the $\delta^{18}O$ values are expressed on VPDB scale.

26. Fig. 1: the expression "water bath" is misleading better to use an expression like "gas thermostat system" or something alike. Clarify whether you used PTFE (brand, type, producer etc.) or Swagelok filters.

Response: "Expression "water bath" is changed to "Thermostat unit" in the revised manuscript.

We used Swagelok filter (Stainless Steel In-Line Particulate Filter, 6 mm Swagelok Tube Fitting, 15 Micron Pore Size) and is corrected in the revised manuscript.

27. Fig 5 and Fig 6: it would be better to use D-δ or Diff-δ instead of Δδ

Response: Diff-δ is used instead of Δδ in the revised manuscript.

28. Fig. 8: Indicate in the figure legend that this is a "... Time course of the evolution of ..." with the specific time resolution.

Response: In the revised manuscript, the mentioned figure number is changed to Figure 9.

Legend is corrected to "Time course of the evolution of soil gas $CO_2$ [ppm], $\delta^{13}C$ and $\delta^{18}O$ in calcareous (a,c,e) and acidic (b,d,f) soils. Data collected continuously over a 12 hour time frame for the calcareous soil and a 14 hour time window with intermittent data collection for the acidic soil.

29. Fig. 9: Indicate in the figure legend that you display "…Daily? averages of CO2 concentration and isotope values in depth profiles…"

Response: In the revised manuscript, the mentioned figure number is changed to Figure 10. Legend is corrected to "Daily average data of soil $CO_2$ [ppm], $\delta^{13}C$ and $\delta^{18}O$ in calcareous (a,b,c) and acidic (d,e,f) soils across soil depth profiles."

Table.1

| Equation | y=a*(b-exp(-c*x)) | | | | |
|---|---|---|---|---|---|
| Standard Error | 0.07468171 | | | | |
| Correlation Coeff.(r) | 0.999941 | | | | |
| Coeff.of Determination (r^2) | 0.99988246 | | | | |
| DOF | 54 | | | | |
| AICC | -294.6349 | | | | |
| Parameters | | | | | |
| Value | Std | Err | Range | (95% | confidence) |
| a | 31.007446 | 0.214984 | 30.576428 | to | 31.438463 |
| b | 0.713759 | 0.002376 | 0.708995 | to | 0.718522 |
| c | 0.000043 | 0 | 0.000042 | to | 0.000043 |
| | | | | | |
| Covariance Matrix | | | | | |
| | a | b | c | | |
| a | 8.286768 | 0.088333 | -0.000018 | | |
| b | 0.088333 | 0.001012 | 0 | | |
| c | -0.000018 | 0 | 0 | | |

Table.2

[revised manuscript text omitted]

Supplementary Table.1 The $\delta^{13}C$ and $\delta^{18}O$ values of the calibration standards used measured against VPDB.

| CO$_2$ standard | $\delta^{13}C$ | $\delta^{18}O$ |
|---|---|---|
| Heavy standard | -4.28 ± 0.03‰ | -9.66 ± 0.06‰ |
| Validation standard | -22.02 ± 0.04‰ | -16.63 ± 0.035‰ |
| Light standard | -39.76 ± 0.04‰ | -23.74 ± 0.035‰ |

---

## Author Response (AR2)

1.  The text on 63-70 would likely lead a casual reader to think that your work was the first to use high-frequency measurements of isotopes in soil gas profiles, but this is not the case. I made a previous comment (not addressed in the response) about the importance of mentioning recent studies that have similarly worked with high-frequency measurements of C and O isotopes in soil profiles, using different analytical techniques. For example, the papers by Jochheim et al. 2018 (10.1002/jpln.201700259), and Bowling et al. 2015 (doi:10.5194/bg-12-5143-2015) are directly related to your topic and should be acknowledged. Also perhaps see Stumpp et al. 2018 (doi:10.2136/vzj2018.05.0096) and papers in that issue.

    Response: First of all, we thank the reviewer for suggesting some relevant work done in this direction. We have referenced those studies in the modified version of our manuscript. We agree on the fact that several studies are already in place using high-frequency measurements of isotopes in soil gas profiles, and is not well addressed in our manuscript. However, we did not come across any work detailing simultaneous measurement of $^{18}O$ and $^{13}C$ in soil derived $CO_2$ using an OA-ICOS, across a depth profile of $0 - 80cm$. We consider our work to be novel in this aspect.

    " **Recently, several high frequency online measurements of $\delta^{13}C$ and $\delta^{18}O$ of soil $CO_2$ and $^2H$, $^{18}O$ of soil water vapor across soil depth profiles were reported by coupling either hydrophobic but gas permeable membranes (installed at different depths in soil) or automated chamber systems with laser spectrometers (Bowling et al., 2015; Jochheim et al., 2018; Stumpp et al., 2018). Such approaches enable detection of vertical concentration profiles, temporal dynamics of soil $CO_2$ concentration and isotopic signature of soil $CO_2$ across different soil layers, thus aiding to identify and quantify various sources of $CO_2$ across the depth profile.** "

2.  I also previously made a comment with respect the "1 hz sampling frequency" that was not understood: "Finally, it should be noted that the useful temporal resolution of the measurements will never actually be 1hz as reported given the Allan variance results." The point here, shown in Figure 3, is that the practical resolution of the measurement cannot be 1 hz because of the high variance in the measured delta values (especially for 18O, ~1.3 per mil) when estimated using 1 hz data. If we assume that precision of say ~0.1 - 0.2 per mil is adequate (obviously, this would depend on the specific study), the useful sampling frequency would be ~10 – 20 seconds (0.05 – 0.1 hz). In fact, your in-situ soil measurements were conducted over 6 minute intervals to allow establishment of steady-state conditions. Please clarify accordingly in the Abstract. This detail is important for readers considering other applications of the method that might demand a higher sampling frequency.

Response: We agree with the reviewer on the fact that at better precision is not achieved at 1hz temporal resolution and that this is clear from the Allan variance results. This was a misunderstanding and is corrected in the manuscript.

"We established a real-time method for measuring soil $CO_2$ concentration, $\delta^{13}C$ and $\delta^{18}O$ values across a soil profile at higher temporal resolutions (0.05 – 0.1 hz) using an Off-Axis Integrated Cavity Output Spectrometer (OA-ICOS)."

*Specific Comments*

1. 30-31: This statement is a truism, as atmospheric $CO_2$ will always diffuse into soil. What is relevant is the degree to which atmospheric $CO_2$ is diluted by soil-respired $CO_2$. There is abundant previous work in this area. Please rephrase.

   Response: We agree that such a rephrased in the modified manuscript.

   "$^{13}C$-$CO_2$ of top soil at the calcareous soil site was found to be reflecting $\delta^{13}C$ values of atmospheric $CO_2$ and $\delta^{13}C$ of top soil $CO_2$ at the acidic soil site was representative of the biological respiratory processes."

2. 31: What is the corollary—were $^{18}O$ values decoupled from soil water at the calcareous site?

   Response: At 80 cm depth in calcareous soil, the $^{18}O$ values were found to be rather enriched relative to the upper soil layers. We rephrased the sentence as follows:

   "$\delta^{18}O$ values of $CO_2$ in both sites reflected the $\delta^{18}O$ of soil water across most of the depth profile, except for the 80 cm depth at the calcareous site where a relative enrichment in $^{18}O$ was observed."

3. 38: "accurate monitoring and modeling of these fluxes are inevitable" I think you mean essential here, rather than inevitable?

Response: Yes, inevitable seems to be too strong for a word in this context. Corrected in the manuscript.

"To understand the prevailing climatic conditions and predict climate change, accurate monitoring and modeling of these fluxes are essential (Barthel et al., 2014; Harwood et al., 1999; Schär et al., 2004)."

4. 38: "Approximately 30 - 35% …" note that the anthropogenic $CO_2$ flux has doubled since this paper was written so this statement is no longer correct.

Response: Agree. This is rectified in the modified manuscript.

"Soil respiration, the $CO_2$ flux released from soil surface to the atmosphere as a result of microbial and root respiration (heterotrophic and autotrophic) is the second largest terrestrial carbon flux (Bond-Lamberty and Thomson, 2010)."

5. 233: replace "inevitable" with "necessary" (that seems to be what you mean?)

Response: Yes, and is changed in the manuscript.

"we found that routine calibration (Correction for concentration-dependent error plus three-point calibration) was necessary for obtaining the required accuracy, in particular under fluctuating $CO_2$ concentrations."

6. 270-277: There is a subtle misinterpretation of your carbonate end member $^{13}C$ values. It is stated "According to Cerling (1984), the distinct oxygen and carbon isotopic composition of soil carbonate depends on the isotopic signature of meteoric water and to the proportion of $C_4$ biomass present at the time of carbonate formation (Cerling, 1984)." Note that pedogenic carbonate $^{13}C$ reflects the $^{13}C$ of the $CO_2$ source of that carbonate (after accounting for fractionation), which is not simply a function of $C_3$ vs. $C_4$ biomass, but rather all of the other myriad factors that determine the $^{13}C$ value of soil $CO_2$. This should be clarified. Then it is stated "$CO_2$ released as a result from carbonates have a distinct $\delta^{13}C$ value close to 0‰ vs. VPDB." Really, the $CO_2$ released from carbonate will have whatever $^{13}C$ value the carbonate had to begin with (assuming complete conversion through bicarbonate to $CO_2$, without fractionation). Note that these in fact have $^{13}C$ values much lower than zero per mil at your site! You have now measured carbonate $^{13}C$ so you can be more precise here.

Response: Agree, as it is clear from our carbonate $^{13}C$ measurements, the major proportion of carbonate in our study site (calcareous) is pedogenic. However, the $^{13}C$ signal of $CO_2$ emanating from geogenic carbonates will have an isotopic signal close to 0‰ vs. VPDB. From our carbonate $^{13}C$ analysis, we get to see that $^{13}C$ signal is approximately close to -6‰ at 80 cm depth and near -9‰ at the upper layers. Since soil at the calcareous site is fluvic Gleysol, the possibility of geogenic carbonates contributing to the $^{13}C$-$CO_2$ signature cannot be neglected. Hence it is probably a mix of biogenic, pedogenic and geogenic carbonates that contribute to the observed $^{13}C$ signature.

"According to Cerling (1984), the distinct oxygen and carbon isotopic composition of soil carbonate depends primarily on the isotopic signature of meteoric water and to the proportion of $C_4$ biomass present at the time of carbonate formation (Cerling, 1984), but also on numerous other factors that determine the $^{13}C$ value of soil $CO_2$. $CO_2$ released as a result from carbonates in calcareous soil site have a distinct $\delta^{13}C$ value of -9.3 (mean value across soil profile 0 - 80 cm depth) (Figure 8(c)), while $CO_2$ released during biological respiratory processes has $\delta^{13}C$ values around -24‰ as observed in the acidic soil (Figure 10 (e)). The $\delta^{13}C$ values of soil $CO_2$ observed in the deepest soil layer in the calcareous soil site most likely indicates the presence of carbonate sources of pedogenic and geologic origin."

7. 283: You measured total inorganic C, not bicarbonate, correct? Is this a typo or something else?

Response: We have measured total Carbon content (that include both organic and inorganic carbon), bulk soil $\delta^{13}C$, carbonate $\delta^{13}C$ & $\delta^{18}O$ values. For measuring carbonate $\delta^{13}C$ & $\delta^{18}O$ values We extracted $CO_2$ from carbonate by treating with phosphoric acid (for details see https://doi.org/10.1016/j.ijms.2006.11.006)

8. 291: Also organic acids (or any other source of H+), which are perhaps most important. Note that acidity generated from $CO_2$ (carbonic acid) will dissolve carbonate to form bicarbonate, but this is a net zero $CO_2$ flux, even though you will observe the $^{13}C$ from the carbonate due to exchange. See for example Zamanian et al. 2016, http://dx.doi.org/10.1016/j.earscirev.2016.03.003

Response: Thanks for the reference, This relevant information is added in the modified manuscript.

"Water content, soil $CO_2$ concentration and presence of organic acids or any other source of $H^+$ are the major factors influencing carbonate weathering, and variations in soil $CO_2$ partial pressure, moisture, temperature, and pH can cause degassing of $CO_2$ which contributes to the soil $CO_2$ efflux (Schindlbacher et al., 2015; Zamanian et al., 2016). $CaCO_3$ solubility in pure $H_2O$ at 25°C is 0.013 $gL^{-1}$, but in weak acids like carbonic acid, the solubility is increased up to five fold (Zamanian et al., 2016). The production of carbonic acid due to $CO_2$ dissolution will convert carbonate to bicarbonates resulting in exchange of carbon atoms between carbonates and dissolved $CO_2$."

9. 328-329: This is another place where relevant recent studies of soil gas isotope dynamics should be cited.

Response: Recent studies of soil gas isotope dynamics are now cited in the modified manuscript.

"Given the fact that laser-based $CO_2$ isotope analyzers are deployed on site in combination with different gas sampling methods like automated chambers systems (Bowling et al., 2015), and hydrophobic gas permeable membranes (Jochheim et al., 2018) for tracing various ecosystem processes, it is important to address this issue."

10. Figure 6: What do the colored dashed lines represent? There is no indication in the legend. Are they some kind of error around the solid colored lines?

Response: Colored dashed lines denote 95% confidence interval. This is corrected in the manuscript.

11. Figure 7: What $CO_2$ mole fractions were used to generate this figure? This seems important in light of Figure 6

Response: $\delta^{13}C$ and $\delta^{18}O$ values corresponding to $CO_2$ concentrations ranging from 400 ppm to 25000 ppm are used to generate the 3-point calibration lines.

[revised manuscript text omitted]